# Comparative Analysis of Carbohydrates, Nucleosides and Amino Acids in Different Parts of *Trichosanthes kirilowii* Maxim. by (Ultra) High-Performance Liquid Chromatography Coupled with Tandem Mass Spectrometry and Evaporative Light Scattering Detector Methods

**DOI:** 10.3390/molecules24071440

**Published:** 2019-04-11

**Authors:** Huang-qin Zhang, Pei Liu, Jin-ao Duan, Ling Dong, Er-xin Shang, Da-wei Qian, Zhen-hua Zhu, Hui-wei Li, Wei-wen Li

**Affiliations:** 1Jiangsu Collaborative Innovation Center of Chinese Medicinal Resources Industrialization, Nanjing University of Chinese Medicine, Nanjing 210023, China; zhqlhw@126.com (H.-q.Z.); shex@sina.com (E.-x.S.); qiandw@njucm.edu.cn (D.-w.Q.); 04040416@163.com (Z.-h.Z.); weijiayoulhw@163.com (H.-w.L.); 2National and Local Collaborative Engineering Center of Chinese Medicinal Resources Industrialization and Formulae Innovative Medicine, Nanjing University of Chinese Medicine, Nanjing 210023, China; 3Jiangsu Key Laboratory for High Technology Research of TCM Formulae, Nanjing University of Chinese Medicine, Nanjing 210023, China; 4Institute of Horticulture, Anhui Academy of Agricultural Sciences, Hefei 230001, China; dlaaas@126.com (L.D.); doulala8404@126.com (W.-w.L.)

**Keywords:** *Trichosanthes kirilowii* Maxim, different parts, carbohydrate, nucleosides, amino acids, resource utilization

## Abstract

*Trichosanthes kirilowii* Maxim. is one of the original plants for traditional Chinese medicines Trichosanthis Fructus, Trichosanthis Semen, Trichosanthis Pericarpium and Trichosanthis Radix. Amino acids, nucleosides and carbohydrates are usually considered to have nutritional value and health-care efficacy. In this study, methods involving high-performance liquid chromatography coupled with evaporative light scattering detector (HPLC-ELSD), UV-visible spectrophotometry and ultra-high-performance liquid chromatography coupled with tandem mass spectrometry (UHPLC-MS/MS) were established for quantifying carbohydrates (fructose, glucose, stachyose, raffinose and polysaccharide), fourteen nucleosides and twenty one amino acids. Moreover, sixty-three samples from nine different parts, including pericarp, seed, fruit pulp, stem, leaf, main root, main root bark, lateral root and lateral root bark of *T. kirilowii* from different cultivated varieties were examined. The established methods were validated with good linearity, precision, repeatability, stability, and recovery. The results showed that the average content of total amino acids in roots (15.39 mg/g) and root barks (16.38 mg/g) were relatively higher than for others. Contents of nucleosides in all parts of *T. kirilowii* were below 1.5 mg/g. For carbohydrates, fruit pulp has a higher content than others for glucose (22.91%), fructose (20.63%) and polysaccharides (27.29%). By using partial least-squared discriminate analysis (PLS-DA), Variable importance in the projection (VIP) plots and analysis of variance (ANOVA) analysis, the characteristic components of the different organs (fruit, stems and leaves, roots) were found. This analysis suggested there were potential medicinal and nutritive health care values in various parts of the *T. kirilowii*, which provided valuable information for the development and utilization of *T. kirilowii*.

## 1. Introduction

*Trichosanthes kirilowii* Maxim. (*T. kirilowii*) is a perennial vine of the Cucurbitaceae family. The fruit, pericarp, seeds and roots of *T. kirilowii* are traditional Chinese medicines recorded in Chinese Pharmacopoeia 2015, named Trichosanthis tructus, Trichosanthis pericarpium, Trichosanthis semen and Trichosanthis radix, respectively. The fruit parts (fruit, pericarp and seeds) of *T. kirilowii* were mainly used for the treatment of cardiovascular diseases and lung diseases in the clinic [1,2,3]. In addition, the root was rich in protein [4,5], which had cytotoxicity to tumor cells [6,7] and hypoglycemic effects [8]. In addition, the leaf, stem, fruit pulp and root bark of *T. kirilowii* were usually regarded as waste, took up valuable land resources and caused environmental problems. Studies have revealed that *T. kirilowii* contains various constituents such as terpenoids [9,10], flavonoids [11], phytosterols [1], amino acids [12], nucleosides, nucleobases and carbohydrates [13]. However, pharmacological reports most often attribute some activities to various extracts rather than specific compounds. The major active compounds of *T. kirilowii* were not clear. Studies have shown that nucleosides, nucleobases and amino acids had clear pharmacological effects on the cardiovascular system [14]. l-arginine was the substrate for endothelial nitric oxide synthesis, and modulated the development of atherosclerotic cardiovascular disease, improving immune function to a healthy level [15]. l-citrulline supplementation increased nitric oxide synthesis, decreased blood pressure, and may increase peripheral blood flow [16]. Cytosine and guanine had the effect of inhibiting platelet aggregation and improving endothelial vascular function [17]. Meanwhile, carbohydrates also had extensive biological effects. Botanical polysaccharides from a wide array of different species of plant are a class of macromolecules that can markedly enhance and activate immune responses [18,19], leading to immunomodulation [20], antitumor activity [21], antiaging activity [22,23] and other therapeutic effects [24].

Considering multiple bioactivities and benefits for human health of these compounds, several analytical methods to detect them have been reported to assess the quality of food or medicinal materials [25,26]. However, nucleosides, nucleobases, amino acids and carbohydrates in medicinal organs (root, pericarp and seed kernels) and non-medicinal organs (fruit pulp, root bark, stem and leaf) of *T. kirilowii* had rarely been investigated. More information about the nutrients in different parts of *T. kirilowii* would, thus, have a significant impact on the efficient use of this valuable natural resource. Thus, it is necessary to establish a fast, convenient, and effective method to clearly characterize and quantify these constituents. On the basis of our previous research [27,28,29], the UHPLC coupled to a triple quadrupole (TQ) MS-MS method to simultaneously detect and quantitate 14 nucleosides and 21 amino acids in different parts of *T. kirilowii* was developed and validated in this study. The high-performance liquid chromatography coupled with an evaporative light scattering detector (HPLC-ELSD) method was established for the detection of four monooligosaccharide, such as fructose, glucose, stachyose and raffinose. At the same time, polysaccharides in samples were determined by UV-visible (UV-Vis) spectrophotometer at 490 nm wavelength. Chemometrics are effective methods for multivariate statistical analysis, while the supervised partial least-squared discriminate analysis (PLS-DA) model and analysis of variance (ANOVA) were employed to find potential markers for the different organs (fruit, stems and leaves, roots) of *T. kirilowii*. The determination of nucleosides, amino acids and carbohydrates in the medicinal parts of *T. kirilowii* can provide theoretical support for the study of the efficacy of medicinal parts. At the same time, the determination of these nutrients can facilitate the development and utilization of non-medicinal organs.

## 2. Results and Discussion

### 2.1. Optimization of Mass Spectrometry and Chromatographic Conditions

Since nucleosides and amino acids have little ultraviolet (UV) absorbance, their detection and accurate quantitation is a challenge with weak retention and poor separation as well as difficult detection using the conventional reversed-phase high-performance liquid chromatography (RP-HPLC)-UV method [30]. To avoid these problems, the UHPLC-MS/MS technique was applied for analysis and identification of nucleosides and amino acids from *T. kirilowii.* In this study, standard solutions of all of the analytes were firstly detected by direct full scan mass spectrometry method in both positive and negative ionization modes. It was observed that all the standards presented higher sensitivity and intensity in positive ion mode than in negative ion mode. Thus, the ESI^+^ mode was adopted in the following experiments. To select a proper transition for the MS/MS detection of the analyte, all the compounds were examined separately in direct infusion mode, and at least two precursor/product ion pairs for each analyte were presented in this study. Then, according to the quantitative results, the highest sensitive and specific ion pairs were selected for the MRM determination. As a result, [M + H]^+^ were considered as the precursor ions for all the target compounds. The most prominent product ions were automatically chosen according to the stability and ion response by MS for further analysis to improve the selectivity and sensitivity in the analysis. The multiple reaction monitoring (MRM) transitions and parameters of 35 compounds applied in the study are listed in Table 1.

Chromatographic parameters were optimized for achieving a higher separation quality of the chromatogram and reducing the analysis time. As for the mobile phase, acetonitrile is known as a polar aprotic solvent with better separation selectivity, elution ability, and peak shape compared to methanol, so acetonitrile was selected as the organic solvent. According to literatures [27,30], the addition of ammonium formate and ammonium acetate to the mobile phase can improve the separation of nucleosides and amino acids. The optimized buffer salt concentration was selected as 5 mmol/L ammonium formate and ammonium acetate in the aqueous phase and 1 mmol/L ammonium formate and ammonium acetate in acetonitrile. Meanwhile, an acidic pH is also used to inhibit solute ionization to improve the peak shape and the sensitivity, so different concentrations of formic acid were added and investigated. The results showed that 0.2% formic acid mixed in the salt mobile phase afforded better peak shapes for most of the analytes. The typical chromatograms of the 35 analytes and sample S1 (Wanlou 7) are presented in Figure 1.

Similarly, for the analysis of mono-oligosaccharides in *T. kirilowii*, according to literatures [26,31], water/acetonitrile was chosen as the preferred mobile phase, and gradient elution was applied during the liquid chromatography process. The typical chromatograms of mono-oligosaccharides and sample S1 (Wanlou 7) are presented in Figure 2.

### 2.2. Method Validation

The developed UHPLC-MS/MS method for quantitation of nucleosides and amino acids, and the HPLC-ELSD method for quantitation of mono-oligosaccharides were validated by determining the linearity, LODs, LOQs, precision, repeatability, stability, and recovery. The results are shown in Table 2. All of the marker substances showed good linearity with the determination coefficients (R^2^) ranging from 0.9908 to 0.9999 within the determination ranges. The analytical precisions for the mixed standard solution were acceptable with an RSD < 4.21%. Stability RSDs of analytes were under 4.22%, indicating the sample solutions were stable for at least two days when stored at 4 °C. Recovery tests showed that the methods were accurate enough for the determination of the analytes in various parts of *T. kirilowii* with overall recoveries between 96.4–104.5%, for which the RSDs < 4.05%. The slope ratio values of the matrix curve to neat solution curve were between 0.94 and 1.07, indicating that the matrix effect on the ionization of analytes was not obvious under these conditions.

### 2.3. Quantitative Analysis of Samples

#### 2.3.1. Distribution Characteristics of Amino Acids and Nucleosides

The established UHPLC-MS/MS method was subsequently applied to analyze 14 nucleosides and 21 amino acids in different parts of *T. kirilowii*. The concentrations of each analyte were calculated via an external standard method. As shown in Appendix A, almost all of the *T. kirilowii* samples were rich in amino acids, despite their contents obviously varying, and the total content of amino acids varied from 0.4449 to 21.29 mg/g among different parts of *T. kirilowii*. As shown in Figure 3, the average content of total amino acids in roots (15.39 mg/g) and root barks (16.38 mg/g) was relatively higher than others, in the order: lateral root barks (17.51 mg/g) > lateral roots (16.15 mg/g) > main root barks (15.27 mg/g) > main roots (14.64 mg/g) > pericarp (7.637 mg/g) > stem (7.177 mg/g) > leaves (7.003 mg/g) > fruit pulp (4.854 mg/g) > seed kernel (2.409 mg/g). The leaves were found to be the most abundant essential amino acid in all parts, and their average content in these investigated samples was 2.551 mg/g, which accounted for more than 35% of total amino acids tested in this study. Next was fruit pulp, whose average essential amino acid content was 1.784 mg/g. In terms of individual compound, as showed in Table 3 and Appendix A, citrulline was found to be the most abundant in all parts except for fruit pulp and seed kernel, and the contents in roots and root barks were more than 9.2 mg/g, which accounted for more than 60% of total amino acids.

Contents of nucleosides in all parts of *T. kirilowii* were below 1.5 mg/g, and only the contents in leaves (1.312 mg/g), lateral root barks (1.255 mg/g) and stem (1.054 mg/g) were above 1mg/g. Next, chemometrics, as effective tools of multivariate statistical analysis, were employed to classify and depict the intrinsic differences of *T. kirilowii* samples.

The results showed that the content of amino acids in the root was higher than other parts, so roots of *T. kirilowii* were good choice of raw material source for amino acids. The amino acid composition and content in the non-medicinal part root bark were similar to those in the root, thus, the non-medicinal part root bark could be a good choice of utilization of amino acids. In addition, the pericarp was also rich in amino acids, especially the citrulline. The leaves of *T. kirilowii* were rich in essential amino acids, suggesting that the leaves could be developed into health products.

#### 2.3.2. Distribution Characteristics of saccharides

The HPLC-ELSD method was applied for mono-oligosaccharides analysis of the samples, and the contents of the four saccharides (fructose, glucose, raffinose and stachyose) were shown in Figure 3, which revealed that the contents of four saccharides in each part of *T. kirilowii* differ from each other. The contents of maltose, mannose and sucrose in all samples were below the limit of detection. In the fruit pulp of *T. kirilowii*, the content of glucose (22.91%), which possessed a high development value, was significantly higher than other parts of the plant, followed by the pericarp (21.77%). Similarly, the fructose was detected mostly in the pericarp (21.04%), followed by the fruit pulp (20.63%). However, the stachyose mainly existed in lateral roots and root barks. Raffinose mainly existed in leaves (1.32%). In addition, the results showed that the polysaccharides in fruit pulp reached up to 27.29%.

The results revealed that in *T. kirilowii,* the monosaccharides and polysaccharides mainly appear to synthetize in genital organs, like fruit pulp and pericarp, while constituents of oligosaccharide may form by dehydration with different kinds of monosaccharides in conducting tissue and vegetative organs, such as stems and roots. Therefore, it can provide guidance for obtaining various saccharide resources from *T. kirilowii*, as it was known to us that saccharides and their derivatives may always be the important components that are responsible for the pharmacological activities of traditional Chinese medicine (TCM). As a non-medicinal part, fruit pulp contained a large amount of fructose, glucose and polysaccharide, suggesting that fruit pulp of *T. kirilowii* might provide a supplementary source of saccharides.

### 2.4. Chemometric Analysis

To further evaluate the variation of amino acids, nucleosides and saccharides in all samples, partial least-squared discriminate analysis (PLS-DA) was performed on the basis of the contents of tested compounds by SIMCA-P 14.1 software (Umetrics AB, Umeå, Sweden). The quality of the PLS-DA model was described by the cross validation parameter Q2, indicating the predictability of the model, and R2Y, which represents the total explained variation for the X matrix [32,33]. Excellent models were obtained when the cumulative values of R2Y and Q2 are above 0.8. Variable importance in the projection (VIP) was used for the selection of potential biomarkers of various organs of *T. kirilowii*. Variables with VIP values larger than 1 were chosen to be more important for the explanation of Y (response). The indexes with VIP values larger than 1 were evaluated with one-way ANOVA for comparison between the groups by SPSS 18.0 (SPSS Inc., Chicago, IL, USA). The *P* value of less than 0.05 was considered statistically significant.

In this study, a supervised PLS-DA model was built to find potential markers for the different organs (fruit, stems and leaves, roots). Clear separation among fruit, stems and leaves, roots groups from score plot of the PLS-DA was easily seen in Figure 4. R2Y and Q2 of the PLS-DA model were 0.958 and 0.944, respectively. The result indicated that the PLS-DA model was good for fitness and prediction. By VIP plot, as shown in Appendix A
Appendix A, nineteen analytes, such as l-citrulline, γ-aminobutyric acid, stachyose, l-arginine, thymidine, 2′-deoxyuridine, l-threonine, l-serine, adenine, 2′-deoxyguanosine, guanosine, fructose, glucose, l-glutamic acid, uridine, inosine, l-tyrosine, *trans*-4-hydroxy-l-proline and 2′-deoxyinosine, with VIP value greater than 1, contributed to the separation of the organs that was detected in the samples. Statistical analysis (ANOVA analysis, *p* < 0.05) of the contents of these nineteen analytes in each organ was conducted. When the content of a compound in an organ was significantly different from the other two groups, the compound was considered to be a characteristic compound of this organ. As showed in Table 4, the characteristic components in fruit were fructose, glucose, serine, threonine, guanosine and uridine, whose contents in fruit were significantly different from other parts. Similarly, the characteristics of stems and leaves were glutamate, hydroxyproline, 2′-deoxyuridine, 2′-deoxyinosine, 2′-deoxyguanosine, inosine and thymidine. Those of the root were stachyose, citrulline and arginine.

## 3. Materials and Methods

### 3.1. Chemicals and Materials

Acetonitrile (HPLC grade), ammonium formate, ammonium acetate and formic acid were all purchased from Merck (Darmstadt, Germany); ultra-pure water was obtained from a Milli-Q water purification system (Millipore, Billerica, MA, USA). The moisture content of the samples was determined using an ADAM PMB-53 automatic moisture meter (ADAM equipment Co. Ltd., Wuhan, China). Other chemicals and reagents were of analytical grade.

Chemical standards of thymidine (1), 2′-deoxyuridine (2), adenine (3), uridine (4), adenosine (5), inosine (8), cytosine (9), 2′-deoxyguanosine (11), cytidine (12), guanosine (13), guanosine 3′5-cyclic monophosphate (14), γ-aminobutyric acid (19), *trans*-4-hydroxy-l-proline (25), l-glutamine (28), l-asparagine (30), l-citrulline (31), l-ornithine (34) and l-cystine (35) were obtained from Sigma (St. Louis, MO, USA). Chemical standards of xanthine (7), guanine (10), l-phenylalanine (15), l-leucine (16), l-tryptophan (17), iso-leucine (18), l-methionine (20), l-valine (21), l-proline (22), l-alanine (24), l-threonine (26), l-glutamic acid (27), l-serine (29), l-arginine (32) and l-lysine (33) were obtained from Huixing Biochemical Reagent Ltd. (Shanghai, China). Reference compounds of 2′-deoxyinosine (6) and l-tyrosine (23) were respectively obtained from Aladdin Reagent Co., Ltd. (Shanghai, China) and Yuanye Bio-Technology Co., Ltd. (Shanghai, China). Maltose (36), fructose (37), raffinose (38) and stachyose (39) were purchased from Spring and Autumn Biological Engineering Co., Ltd. (Nanjing, China). Mannose (40), glucose (41), sucrose (42), and glucuronic acid were purchased from Solarbio Science&Technology Co., Ltd. (Beijing, China). The purity of each compound was determined to be over 98% by HPLC analysis. Their chemical structures are shown in Appendix A.

### 3.2. Plant Materials

Samples of *T. kirilowii* were collected from the Breeding and Demonstration Base of Gualou Varieties in Changfeng County, Hefei, Anhui Province, China, in November 2017. Seven different planting varieties were collected: Wanlou 7 (S1), Wanlou 4-11 (S2), Wanlou 17 (S3), Wanlou 9 (S4), Wanlou 16 (S5), Wanlou 8 (S6), and Wanlou 7-8 (S7). These samples were all collected during the seed maturity period, except for Wanlou 9 (S4), which was obtained during the mature period. The plants were authenticated by Prof. Dong Ling as *T. kirilowii*. The voucher specimens were deposited in the Jiangsu Collaborative Innovation Center of Chinese Medicinal Resources Industrialization, China. After the samples were collected, the plants were divided into main root (A), lateral root (B), lateral root bark (C), main root bark (D), stem (E), leaf (F), pericarp (G), fruit pulp (H) and seed kernels (I). The pieces were freeze-dried and then pulverized to provide homogeneous powders (40 mesh) for extraction.

### 3.3. Preparation of Standard Solutions

#### 3.3.1. Standard Solution Preparation of Nucleosides and Amino Acids

A mixed standard stock solution containing the reference compounds **1**–**35** was prepared by dissolving them in water/methanol (9:1, *v*/*v*), and the concentrations of these analytes were as follows: 41.7, 47.0, 45.3, 43.7, 47.0, 40.7, 3.60, 41.0, 46.0, 6.20, 40.3, 36.7, 42.0, 38.0, 51.3, 36.7, 51.0, 43.7, 34.0, 42.3, 35.3, 37.7, 35.0, 40.7, 52.7, 43.3, 33.7, 39.0, 45.0, 67.0, 66.7, 35.0, 35.7, 6.50, 40.7 µg/mL, respectively. Working standard solutions for calibration curves were prepared by diluting the mixed standard solution with 10% methanol at different concentrations.

#### 3.3.2. Standard Solution Preparation of Saccharides

The mixed standard stock solution containing the reference compounds **36**–**42** was prepared with ultru-water at concentrations of 10.8, 10.3, 10.3, 9.30, 9.58, 9.96 and 10.7 mg/mL, then diluted to different concentrations. All of the solutions were stored in a refrigerator at 4 °C until use and filtered through a 0.22 µm cellulose membrane before analysis.

For determination of polysaccharide, glucose was accurately weighed, dissolved in deionized water, and the concentration for glucose was 50.04 μg·mL^−1^, then diluted to different concentrations.

### 3.4. Preparation of Sample Solutions

#### 3.4.1. Extraction of Nucleosides, Amino Acids and Mono-oligosaccharide

The dried powder (1.0 g) of samples, which was accurately weighed, was put into a 50 mL glass-stoppered conical flask and 20 mL ultrapure water was added. After resting for one hour and accurate weighing, ultrasonication (40 kHz) was performed at room temperature for one hour. Afterwards solvent was added if there was any weight loss. After centrifugation (13,000 r/min, 10 min) and filtration (0.22 µm membrane filter), the supernatant was stored (4 °C) at a sample plate before injection into the UHPLC and HPLC systems for analysis. The samples of pericarp and fruit pulp were diluted ten times with ultrapure water for determination of mono-oligosaccharide.

#### 3.4.2. Extraction of Polysaccharide

The dried powder (1 g, 40 mesh) from different tissues of *T. kirilowii*, which was weighed accurately, was put into a 50mL conical flask with a stopper and 20 mL 80% ethanol was added. After resting for 1 h and accurate weighing, ultrasonication (40 kHz) was performed at room temperature for 30 min. This process was repeated twice. After filtration, the filter residue was washed with 80% ethanol and put into a 50 mL conical flask with a stopper, and 25 mL water was added. After accurate weighing, the conical flask was placed in the water bath at 100 °C for 2 h; afterwards the same solvent was added to compensate for the weight lost during extraction. After centrifugation (3000 rpm, 10 min), the supernatant was diluted and all of the sample solutions were stored at 4 °C before analysis.

### 3.5. Chromatographic Conditions and Instrumentation

#### 3.5.1. Analysis for Nucleosides and Amino Acids

Chromatographic analysis was performed on a Waters Acquity UHPLC system (Waters, Corp., Milford, MA, USA) consisting of a binary pump solvent management system, an online degasser, and an autosampler. An Acquity UHPLC BEH Amide (100 mm × 2.1 mm, 1.7 µm) column was applied for analyses. The mobile phase was composed of A (5 mM ammonium formate and ammonium acetate, 0.2% formic acid) and B (acetonitrile with 1 mM ammonium formate, ammonium acetate, and 0.2% formic acid) with a gradient elution: 0–3 min, 10% A; 3–9 min, 10–18% A; 9–15 min, 18–20% A; 15–16 min, 20–46% A; 16–18 min, 46% A. The flow rate was 0.4 mL/min. The column temperature was conditioned at 30 °C, and the injection volume was 2 µL.

Mass spectrometry detection was performed using an AB SCIEX Triple Quad 6500 plus (AB SCIEX Corp., Massachusetts, MA, USA) equipped with an electrospray ionization source (ESI). The ESI-MS spectra were acquired in both positive ion multiple reaction monitoring (MRM) mode. The conditions of MS analysis were designed as follows: the capillary voltage at 5.5 kV, the desolvation gas flow rate set to 1000 L/h at a temperature of 550 °C, the cone gas flow rate set at 50 L/h, and the source temperature was 150 °C. The declustering potential (DP) and collision energy (CE) were set to match the MRM of each marker. The dwell time was automatically set by the MultiQuant 3.0.2 software (AB SCIEX Corp., Massachusetts, MA, USA).

#### 3.5.2. Analysis for Mono-oligosaccharide

The HPLC-ELSD conditions were used for the determination of the mono-oligosaccharide, a Waters Alliance 2695 liquid chromatograph system (Waters, Milford, MA, USA) equipped with a Waters 2424 evaporative light-scattering detector (Waters, Milford, MA, USA) was used. The chromatographic separations were performed over a Prevail Carbohydrate ES (250 mm × 4.6 mm, 5 µm) column at a column temperature of 25 °C. The analytes were eluted with a mixture of acetonitrile (mobile phase A) and water (mobile phase B) at a flow rate of 1.0 mL/min. The elution conditions were as follows: 0–7 min, 25% D; 7–17 min, 25–45% D; 17–19 min, 45–50% D; 19–21 min, 25–50% D; 21–25 min, 25–25% D. The drift tube temperature of the ELSD was set at 50 °C, and using nitrogen as the carrier gas at a flow rate set at 2.5 L/min, the gain value was 10, and the injection volume was 10 µL for analysis.

#### 3.5.3. Analysis for Polysaccharides

The polysaccharides were determined as glucose with hydrolyzing polysaccharides into glucose monomer, which based on the color reaction of polysaccharides and their derivatives with phenol and concentrated sulfuric acid [29,34,35]. 1.0 mL of each concentration of glucose standard solution were added into different dry glass test tubes respectively. Meanwhile, place 1 mL water into another test tube as a control. 2.0 mL of 5% phenol solution and 7 mL of concentrated sulfuric acid were added to all the tubes. After being mixed evenly, tubes were placed in the boiling water for 15 min. After cooling, the absorbance was measured at 490 nm wavelength. The calibration curve was carried out with glucose solution concentration. The samples were also treated with 5% phenol solution and sulfuric acid to determine the polysaccharides. Then polysaccharides in samples were determined by UV spectrophotometer at 490 nm wavelength.

### 3.6. Analytical Method Validation

The linearity was constructed by preparing a series of concentrations of standard solution with at least five appropriate concentrations in duplicate. The working solution for calibration use was diluted with the corresponding solvent to a series of concentrations. The LOD and LOQ for each analyte were acquired at the signal-to-noise ratio (S/N) of 3 and 10, respectively. The peak height divided by the background noise value was calculated as the S/N.

To evaluate the method, assay precision was assessed by successively analyzing six injections of mixed standard solutions (high, medium, and low—the highest, middle, and lowest concentration of linearity). Six independent sample solutions (sample S1, all the parts of *T. kirilowii*) were analyzed to confirm the repeatability of the method, and one of the sample solutions was analyzed at 0, 2, 4, 8, 12, 24, 36, and 48 h, respectively, to evaluate the stability of the solution. A recovery test was used to evaluate the accuracy of the method. This was performed by adding the corresponding marker compounds with high (120% of the known amounts), middle (same as the known amounts), and low (80% of the known amounts) levels to an accurately weighed sample (0.5 g) analyzed previously. The spiked samples were then extracted, processed, and quantified as above. Triplicate experiments were performed at each level. The average recoveries were estimated by the formula: recovery (%) = [(amount found − original amount)/amount added] × 100.

The matrix effect was evaluated using the slope ratio of calibration curves of standards in solvent and matrix-matched solutions. The matrix does not suppress or enhance the response of the MS when the slope ratio was 1, otherwise denote ionization suppression (<1) or enhancement (>1).

### 3.7. Sample Determination

All of the samples were prepared according to Section 3.4, and determined thereafter according to the Section 3.5 chromatographic conditions and instrumentation. Quantification of nucleosides and amino acids were calculated via an external standard method. Quantification of mono-oligosaccharide performed on the basis of linear calibration plots, with the logarithm of peak areas versus the logarithm of corresponding concentration. Quantification of polysaccharides was performed on the basis of linear calibration plots of the absorbance versus the corresponding concentration.

### 3.8. Data Processing and Statistical Analysis

The raw data were processed with MultiQuant 3.0.2, SIMCA-P 14.1 and SPSS 22.0 software. All data were normalized and the resultant data matrices were introduced to SIMCA-P 14.1 software (Umetrics AB, Umeå, Sweden) for principle component analysis (PCA), partial least-squared discriminate analysis (PLS-DA) and SPSS 18.0 (SPSS Inc., Chicago, IL) for ANOVA analysis.

## 4. Conclusions

Within the aim of exploring the nutrients in *T. kirilowii* and evaluating its quality, rapid and reliable UHPLC-TQ-MS and HPLC-ELSD analysis was established to determine the potential nutritional compounds, including nucleosides, nucleobases amino acids and saccharides, in different parts of *T. kirilowii*. The data showed that there were remarkable differences in the distribution of the nucleosides, amino acids and saccharides. There were many amino acids in *T. kirilowii* roots, especially citrulline, and they could be extracted and enriched to prepare natural antioxidants, medicines, or health products. The pericarp and fruit pulp could also be used as a source of saccharides. The research could provide the theoretical basis and scientific evidence for comprehensive development and utilization of *T. kirilowii* resources. In addition, the established UV-Vis Spectrophotometry, HPLC-ELSD and UHPLC-MS/MS methods can be applied in future research of *T. kirilowii*.

## Figures and Tables

**Figure 1 molecules-24-01440-f001:**
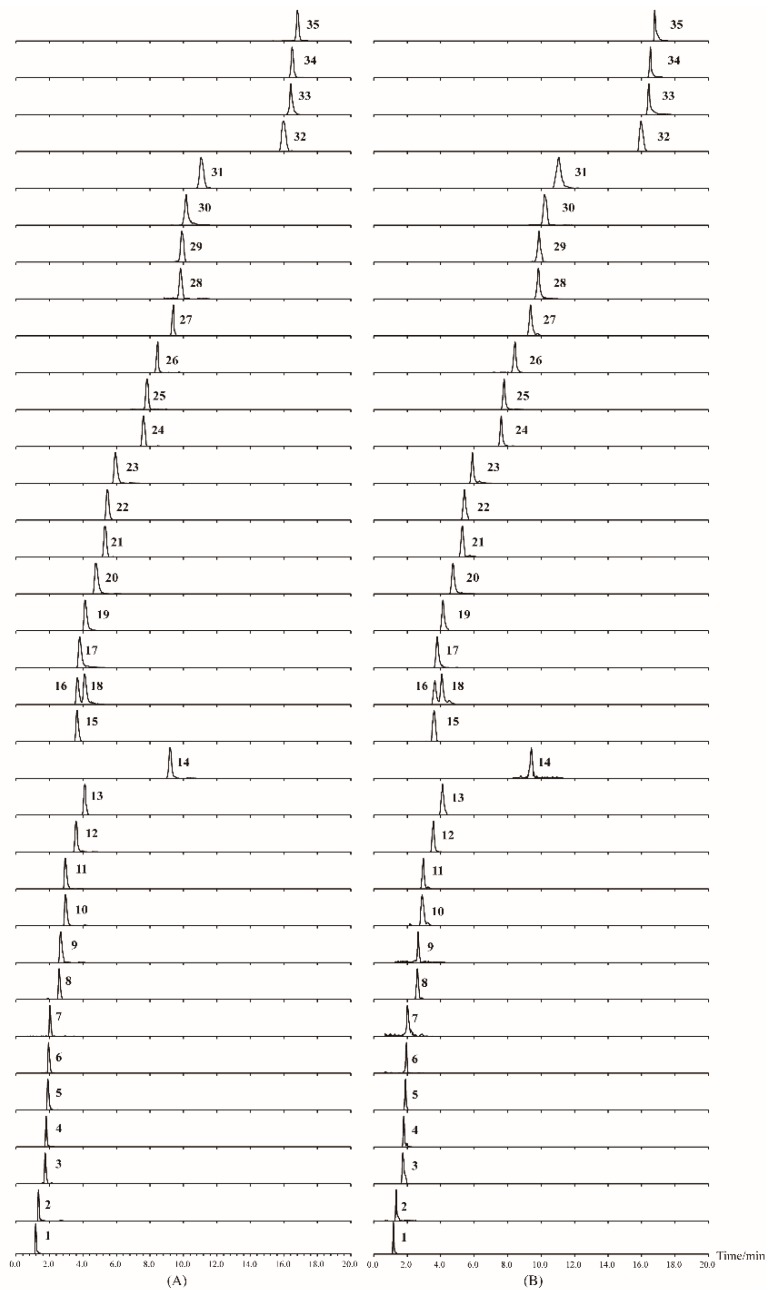
UHPLC-MS-MS chromatograms of mixed standards (**A**) and sample (**B**) for the 35 analytes in this study. The analytes numbers **1**–**35** are consistent with those in Table 1.

**Figure 2 molecules-24-01440-f002:**
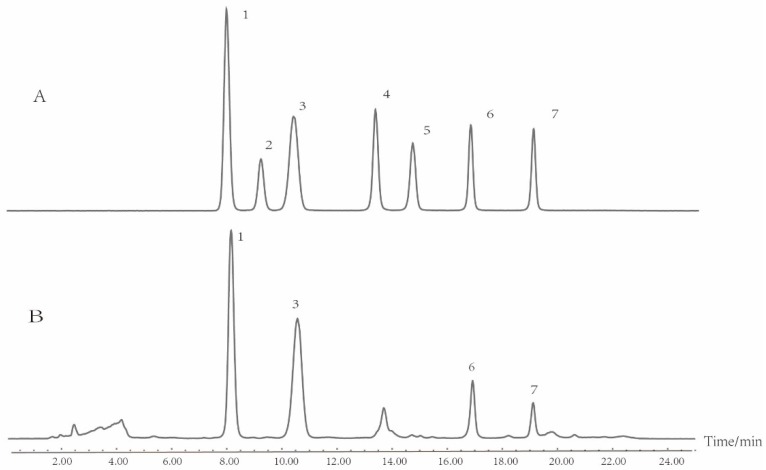
HPLC-ELSD chromatogram of mono-oligosaccharides in *T. kirilowii*. (**A**) Mixed reference; (**B**) leaves of sample S1 (Wanlou 7); (1) fructose; (2) mannose; (3) glucose; (4) sucrose; (5) maltose; (6) raffinose; (7) stachyose.

**Figure 3 molecules-24-01440-f003:**
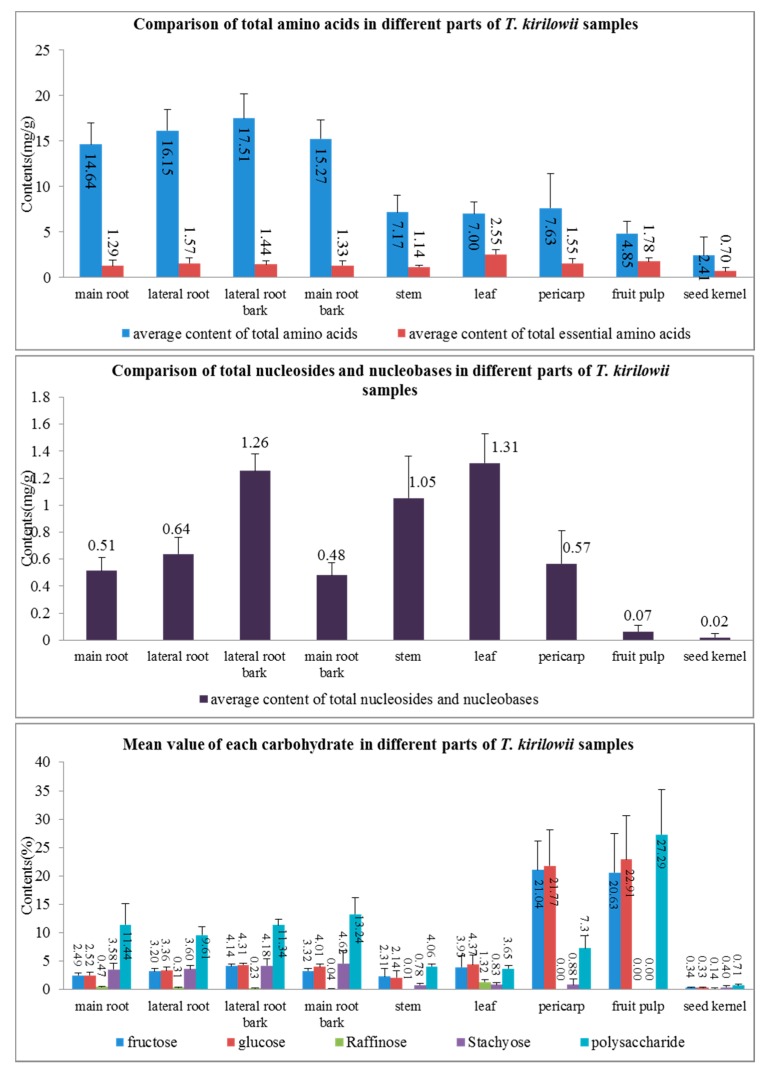
Comparison of nutrients in different *T. kirilowii* samples.

**Figure 4 molecules-24-01440-f004:**
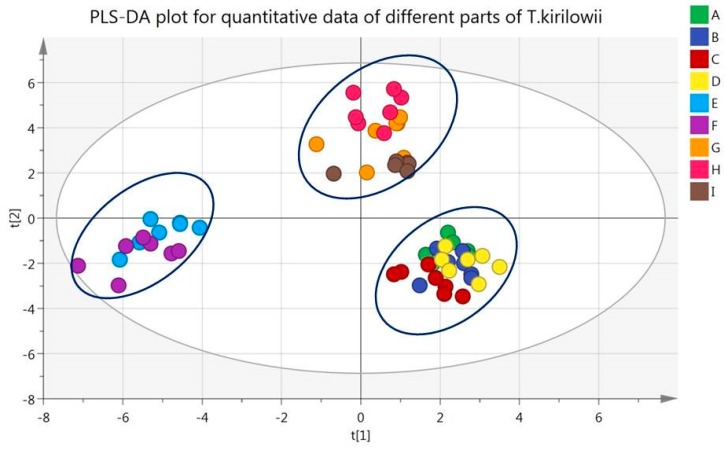
PLS-DA plot of *T. kirilowii* samples. main root (**A**), lateral root (**B**), lateral root bark (**C**), main root bark (**D**), stem (**E**), leaf (**F**), pericarp (**G**), fruit pulp (**H**) and seed kernels (**I**).

**Table 1 molecules-24-01440-t001:** Precursor/product ion pairs and parameters for SIM/MRM of compounds used in this study.

No.	Compound	t_R_ (min)	Precursor Ion (*m*/*z*)	Product Ion ^a^ (*m*/*z*)	DP ^b^ (V)	CE ^b^ (V)	CXP ^b^ (eV)
**1**	Thymidine	1.18	243.052	110.0,117.0,**126.9**	11	13	14
**2**	2′-Deoxyuridine	1.35	229.040	96.0,**112.9**,117.0	11	13	12
**3**	Adenine	1.75	136.077	**92.0**,94.0,109.1	1	41	40
**4**	Uridine	1.80	245.119	**113.0**,115.0,171.0	21	13	12
**5**	Adenosine	1.90	268.023	92.0,119.1,**136.1**	31	23	14
**6**	2′-Deoxyinosine	1.95	253.024	110.0,118.9,**136.9**	11	11	16
**7**	Xanthine	2.02	153.045	80.9,**109.9**,134.9	91	25	10
**8**	Inosine	2.62	269.017	109.9,118.9,**137.0**	16	13	16
**9**	Cytosine	2.68	112.041	52.0,67.0,**69.0**	46	23	6
**10**	Guanine	2.95	152.040	53.0,**109.9**	111	27	12
**11**	2’-Deoxyguanosine	2.95	268.066	110.0,135.0,**152.0**	11	13	16
**12**	Cytidine	3.57	244.161	93.9,94.9,**112.0**	31	15	12
**13**	Guanosine	4.11	284.056	110.1,135.0,**152.0**	1	17	16
**14**	guanosine 3′5-cyclic monophosphate	9.42	345.954	135.0,**152.0**	96	25	18
**15**	l-Phenylalanine	3.66	166.097	77.1,103.0,**120.1**	1	17	14
**16**	l-Leucine	3.67	132.069	55.1,69.0,**86.2**	11	13	10
**17**	l-Tryptophan	3.80	205.089	118.1,143.0,**146.1**	26	25	16
**18**	Isoleucine	4.09	132.069	55.1,69.0,**86.2**	11	13	10
**19**	γ-Aminobutyric acid	4.14	104.022	68.5,69.0,**87.1**	1	17	52
**20**	l-Methionine	4.78	150.117	56.0,61.0,**104.0**	36	15	12
**21**	l-Valine	5.31	118.125	55.0,55.1,**72.0**	1	13	8
**22**	l-Proline	5.43	116.167	68.0,**68.1**,70.0	21	37	8
**23**	l-Tyrosine	5.92	182.097	91.0,123.0,**136.1**	41	19	16
**24**	l-Alanine	7.62	89.988	**55.0**,57.2	51	29	26
**25**	*trans*-4-hydroxy-l-proline	7.79	132.038	68.0,**86.0**	71	19	10
**26**	l-Threonine	8.44	120.128	56.0,**74.0**,102.0	16	13	8
**27**	l-Glutamic acid	9.40	148.051	**84.1**,102.0	66	21	10
**28**	l-Glutamine	9.82	147.072	56.0,**84.1**	51	23	10
**29**	l-Serine	9.90	105.991	**60.0**,70.0	51	15	8
**30**	l-Asparagine	10.18	133.111	73.0,74.0,**87.1**	1	19	32
**31**	l-Citrulline	11.06	176.104	**70.0**,113.0	26	29	8
**32**	l-Arginine	15.98	175.110	**70.1**,116.1	86	27	8
**33**	l-Lysine	16.44	147.110	56.1,**84.0**	61	23	10
**34**	l-Ornithine	16.53	133.082	**70.0**,73.1	51	23	8
**35**	l-Cystine	16.81	240.943	**74.0**,98.9	31	39	10

^a^ Most abundant product ion, with the quantification ion being bolded. ^b^ DP declustering potential, CE collision energy, CXP cell exit potential.

**Table 2 molecules-24-01440-t002:** Linear regression, LOD, LOQ and precision, stability, repeatability, recovery of the analytes.

No.	Regression Equation	R^2^	Linear Range (ng/mL)	LOD	LOQ	Precision (RSD, %)		Stability (RSD, %)	Repeatability (RSD, %)	Recovery	
				(ng/mL)	(ng/mL)	Intraday	Interday			Mean, %	RSD, %
**1**	y = 2,824,247.9x + 5,954,683.1	0.9978	2.500–4.167 × 10^4^	0.13	0.34	2.32	3.02	3.42	2.42	101.3	2.11
**2**	y = 922,961x + 568,615	0.9986	5.100–5.875 × 10^3^	0.58	1.7	1.32	2.42	3.43	4.23	98.0	3.01
**3**	y = 1,486,176.5x + 632,070.7	0.9968	2.700–1.130 × 10^4^	0.11	0.34	0.92	2.23	4.01	3.92	103.3	3.07
**4**	y = 864,935x + 584,181	0.9987	2.600–4.300 × 10^4^	0.13	0.86	2.16	3.12	4.21	4.64	97.4	4.05
**5**	y = 12,340,559.2x + 13,971,574.7	0.9971	5.300–2.300 × 10^4^	0.46	2.4	2.53	3.24	3.98	3.13	104.2	3.75
**6**	y = 4,394,306.8x + 3,947,119.7	0.9987	4.000–2.001 × 10^4^	0.39	1.5	3.04	3.52	4.06	4.14	96.4	3.13
**7**	y = 327,479x + 18,539	0.9988	14.00–3.603 × 10^3^	1.4	6.9	2.91	3.12	3.42	4.01	97.4	3.37
**8**	y = 4,095,340.6x + 4,855,604.6	0.9982	5.200–4.100 × 10^4^	0.84	2.9	3.43	3.32	2.94	3.54	103.2	2.95
**9**	y = 439,258x + 879,574	0.9939	10.10–4.603 × 10^4^	0.91	2.7	1.93	2.42	2.95	3.24	102.3	2.82
**10**	y = 17,063,239.4x + 149,593.2	0.9974	0.7000–1.900 × 10^2^	0.018	0.039	2.82	3.02	3.12	3.32	99.4	3.21
**11**	y = 14,470,625.6x + 1,549,987.5	0.9937	4.000–2.500 × 10^3^	0.68	2.5	3.03	2.94	2.84	3.23	102.4	2.94
**12**	y = 7,629,191.3x + 3,135,983.9	0.9911	2.000–3.600 × 10^4^	0.21	0.36	2.95	3.13	3.65	3.64	103.5	3.53
**13**	y = 4,083,739.7x + 7,735,801.6	0.9921	2.500–1.260 × 10^5^	0.13	0.37	3.54	3.74	3.96	3.44	96.4	2.95
**14**	y = 2,948,202.5x + 1,270,029.0	0.9991	4.600–1.140 × 10^5^	0.46	1.1	3.05	3.54	2.96	3.04	102.4	3.05
**15**	y = 7,971,028.9x + 2,483,617.9	0.9999	6.000–5.101 × 10^4^	0.51	2.0	2.95	3.03	3.32	3.54	104.5	3.41
**16**	y = 4,223,984.3x + 555,633.9	0.9989	8.000–3.603 × 10^4^	0.83	3.6	3.03	3.21	3.33	2.63	103.6	2.94
**17**	y = 4,587,940.9x – 483,834.6	0.9997	12.10–5.102 × 10^4^	0.50	1.3	3.09	3.86	4.05	3.53	96.4	3.53
**18**	y = 4,304,705.8 x + 10,445.9	0.9996	11.00–4.300 × 10^4^	0.42	1.4	4.04	3.95	4.22	3.95	102.4	3.89
**19**	y = 960,985x + 794,899	0.9971	8.000–1.020 × 10^5^	0.30	1.0	3.91	2.94	4.02	3.84	103.5	4.04
**20**	y = 1,828,288.2x – 155,615.5	0.9992	5.000–4.201 × 10^4^	0.53	1.4	3.53	4.02	3.95	4.23	98.3	3.94
**21**	y = 2,631,959.6x + 231,832.9	0.9998	8.100–3.500 × 10^4^	0.35	1.1	3.94	3.89	3.52	3.56	104.2	4.12
**22**	y = 115,788x + 44,801	0.9941	70.10–3.702 × 10^4^	3.7	11	3.53	2.94	3.45	4.05	102.6	3.64
**23**	y = 1,263,773.8x + 124,249.7	0.9996	30.10–3.502 × 10^4^	3.0	9.7	4.03	3.52	3.89	3.51	97.4	3.52
**24**	y = 195,275.9x + 46,906.5	0.9988	1.588×10^2^–2.033×10^4^	15	41	4.21	3.81	3.52	4.02	103.5	4.01
**25**	y = 1,557,660.4x – 664,389.7	0.9974	25.00–5.310 × 10^4^	5.4	12	3.85	3.53	3.90	4.13	102.4	3.89
**26**	y = 550,153x – 170,668	0.9978	20.20–4.301 × 10^4^	0.43	1.4	3.31	2.53	2.95	3.69	101.5	3.64
**27**	y = 1,101,798.4x – 441,695.7	0.9955	31.00–3.370 × 10^4^	0.39	1.2	3.21	3.32	3.13	4.01	97.5	4.64
**28**	y = 2,056,201.7x – 1,509,290.6	0.9932	19.00–3.903 × 10^4^	0.73	2.0	3.53	2.95	3.34	2.95	103.5	2.95
**29**	y = 441,570x + 48,517	0.9997	80.10–4.500 × 10^4^	0.48	1.4	3.63	2.94	3.52	2.53	98.4	3.04
**30**	y = 261,545x – 89,819	0.9994	1.300 × 10^2^–6.700 × 10^4^	1.9	6.7	2.98	3.02	3.32	3.52	102.5	4.04
**31**	y = 1,501,631.2x – 976,172.6	0.9995	16.00–6.602 × 10^4^	0.67	1.7	3.02	2.94	3.12	3.14	100.4	3.53
**32**	y=3,645,684x – 98,187	0.9989	1.360 × 10^2^–3.502 × 10^4^	0.97	2.7	2.79	3.10	2.93	4.02	97.2	2.99
**33**	y = 506,624x – 224,402	0.9922	1.300 × 10^2^–1.781 × 10^4^	0.65	1.8	3.02	2.95	3.04	4.04	99.4	3.02
**34**	y = 255,220x + 20,740	0.9912	60.10–1.600 × 10^3^	0.64	1.6	2.94	3.04	3.53	3.79	101.4	3.53
**35**	y = 300,359x – 36,234	0.9908	4.100–4.001 × 10^4^	0.57	2.0	3.32	3.63	4.04	3.54	104.4	4.14
**36**	y = 1.3913x + 3.9365	0.9976	2.810 × 10^4^–2.160 × 10^7^	3.5 × 10^3^	14 × 10^3^	1.73	2.12	2.42	2.67	102.4	3.04
**37**	y = 1.3796x + 4.2746	0.9953	5.350 × 10^4^–4.108 × 10^7^	6.7 × 10^3^	13 × 10^3^	1.89	2.17	2.75	2.32	98.7	3.02
**38**	y = 1.4068x + 3.8994	0.9988	2.680 × 10^4^–2.060 × 10^7^	3.4 × 10^3^	13 × 10^3^	1.90	2.42	2.48	3.03	101.4	2.97
**39**	y = 1.3851x + 5.4207	0.9964	2.420 × 10^4^–4.650 × 10^7^	3.0 × 10^3^	12 × 10^3^	2.43	2.55	3.14	2.96	98.4	3.08
**40**	y = 1.4024x + 1.8305	0.9986	9.980 × 10^4^–5.474 × 10^7^	12 × 10^3^	25 × 10^3^	2.11	2.53	2.16	2.75	101.9	3.25
**41**	y = 1.4189x + 5.0747	0.9977	5.190 × 10^4^–4.980 × 10^7^	6.5 × 10^3^	13 × 10^3^	1.83	2.43	2.43	2.31	99.9	2.94
**42**	y = 1.4088x + 4.1731	0.9988	2.800 × 10^4^–2.148 × 10^7^	3.5 × 10^3^	14 × 10^3^	1.69	1.93	2.31	2.01	100.4	2.88

**Table 3 molecules-24-01440-t003:** Proportion of the relatively abundant analytes in total amino acids (%).

Analytes	Different Parts of *T. kirilowii*								
	Main Root	Lateral Root	Lateral Root Bark	Main Root Bark	Stem	Leaf	Pericarp	Fruit Pulp	Seed Kernel
glu	0.97	0.68	1.36	1.53	9.56	11.65	0.51	6.38	18.59
lys	3.44	3.20	3.13	3.24	6.09	10.76	7.90	11.67	10.80
gln	4.01	4.33	2.96	2.74	14.70	13.64	8.88	4.79	2.27
ser	0.10	0.12	0.05	0.02	0.31	0.26	3.31	4.18	2.07
cit	63.20	62.13	65.46	62.65	25.64	13.92	40.54	8.57	2.54
orni	1.23	1.55	1.34	1.36	0.91	0.81	0.41	0.84	13.75
cysti	0.01	0.01	0.01	0.01	0.07	0.06	0.03	0.39	0.08
asn	0.48	0.43	0.65	0.50	0.29	2.21	1.95	3.27	2.72
GABA	5.14	5.68	5.84	6.72	6.02	4.54	4.67	4.97	2.77
leu	1.17	1.60	1.01	1.10	1.45	4.29	1.88	5.31	3.22
ile	1.20	1.49	1.22	1.14	2.05	5.70	2.38	5.64	3.31
met	0.61	0.63	0.56	0.58	0.60	1.78	1.16	1.51	1.56
phe	0.35	0.33	0.30	0.32	0.58	1.87	0.96	2.97	2.10
pro	0.77	0.84	1.22	1.08	17.86	5.08	3.03	6.15	3.94
thr	0.00	0.00	0.00	0.00	0.01	0.02	1.74	1.68	1.95
try	0.49	0.63	0.25	0.49	0.72	0.97	0.41	1.12	1.15
tyr	0.71	0.82	1.00	0.67	0.07	0.28	4.59	8.34	3.51
val	1.54	1.80	1.77	1.85	4.41	11.05	3.85	6.85	4.97
hpro	0.01	0.01	0.01	0.01	0.09	0.04	0.03	0.02	0.03
ala	0.49	0.56	0.83	0.44	2.55	4.11	1.44	3.02	1.07
arg	14.08	13.15	11.05	13.54	6.12	7.00	10.37	12.35	17.62

Note: glu, l-glutamic acid; lys, l-lysine; gln, l-glutamine; ser, l-serine; cit, l-citrulline; orni, l-ornithine; cysti, l-cystine; asn, l-asparagine; GABA, γ-aminobutyric acid; leu, l-leucine; ile, iso-leucine; met, l-methionine; phe, l-phenylalanine; pro, l-proline; thr, l-threonine; try, l-tryptophan; tyr, l-tyrosine; val, l-valine; hprp, *trans*-4-hydroxy-l-proline; ala, l-alanine; arg, l-arginine

**Table 4 molecules-24-01440-t004:** The possible characteristic components (VIP > 1) in different organs of *T. kirilowii* (%).

Compounds	Roots	Fruits	Stems and Leaves
l-Citrulline	**1.008 ± 0.2129 (2,3) ^a^**	0.1191 ± 0.0791 (1)	0.1406 ± 0.0918 (1)
γ-Aminobutyric acid	0.0930 ± 0.0143 (2,3)	0.0221 ± 0.0168 (1,3)	0.0375 ± 0.0198 (1,2)
Stachyose	**3.996 ± 1.404 (2,3) ^b^**	0.4293 ± 0.6762 (1)	0.8090 ± 0.3251 (1)
l-Arginine	**0.2047 ± 0.0495 (2,3)**	0.0605 ± 0.0500 (1)	0.0465 ± 0.0159 (1)
Thymidine	(4.284 ± 3.239) × 10^−4^ (3)	(4.047 ± 6.647) × 10^−4^ (3)	**0.0084 ± 0.0014 (1,2) ^a^**
2′-Deoxyuridine	(1.767 ± 1.232) × 10^− 4^ (3)	(1.177 ± 1.761) × 10^−4^ (3)	**0.0028 ± 0.0005 (1,2)**
l-Threonine	(1.944 ± 0.9308) × 10^−5^ (2)	**0.0087 ± 0.0064 (1,3) ^a^**	(8.286 ± 2.098) × 10^−5^ (2)
l-Serine	0.0011 ± 0.0010 (2)	**0.0168 ± 0.0126 (1,3)**	0.0020 ± 0.0018 (2)
Adenine	0.0158 ± 0.0041 (2,3)	0.0041 ± 0.0043 (1,3)	0.0089 ± 0.0066 (1,2)
2′-Deoxyguanosine	(1.744 ± 1.231) × 10^−4^ (3)	(2.026 ± 1.335) × 10^−4^ (3)	**0.0045 ± 0.0030 (1,2)**
Guanosine	0.0112 ± 0.0063 (2)	**0.0018 ± 0.0025 (1,3)**	0.0107 ± 0.0067 (2)
fructose	3.288 ± 0.716 (2)	**14.00 ± 10.94 (1,3)**	3.132 ± 1.851 (2)
glucose	3.550 ± 0.8528 (2)	**15.00 ± 11.95 (1,3)**	3.255 ± 1.826 (2)
l-Glutamic acid	0.0181 ± 0.0095 (3)	0.0265 ± 0.0030 (3)	**0.0751 ± 0.0244 (1,2)**
Uridine	0.0433 ± 0.0251 (2)	**0.0112 ± 0.0175 (1,3)**	0.0508 ± 0.0227 (2)
Inosine	(7.932 ± 6.414) × 10^−5^ (3)	0.0017 ± 0.0006 (3)	**0.0237 ± 0.0191 (1,2)**
l-Tyrosine	0.0128 ± 0.0060 (2,3)	0.0280 ± 0.0202 (1,3)	0.0012 ± 0.0002 (1,2)
*trans*-4-hydroxy-l-proline	(1.029 ± 0.4785) × 10^−5^ (3)	(7.359 ± 5.911) × 10^−6^ (3)	**(2.714 ± 1.556) × 10^−5^ (1,2)**
2′-Deoxyinosine	(1.127 ± 0.9120) × 10^−6^ (3)	(3.998 ± 1.556) × 10^−5^ (3)	**0.0011 ± 0001 (1,2)**

The number 1 in parentheses represents the root, 2 represents fruit, 3 represents stems and leaves. ^a^ (2,3) represents that the content of citrulline in roots was significantly different from that in fruits and stems and leaves. (1,3) represents that the content of analyte in fruits was significantly different from that in roots and stems and leaves. (1,2) represents that the content of analyte in stems and leaves was significantly different from that in roots and fruits. *p* < 0.05. ^b^ The content of the component significantly differed from the other two groups, with the potential markers being bolded

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
