# Peer review of "Comparative Analysis of Carbohydrates, Nucleosides and Amino Acids in Different Parts of Trichosanthes kirilowii Maxim. by (Ultra) High-Performance Liquid Chromatography Coupled with Tandem Mass Spectrometry and Evaporative Light Scattering Detector Methods"

_molecules, 2019, doi:10.3390/molecules24071440_

Round 1
Reviewer 1 Report
First I wanted to highlight the important work that have been done for this study.
Concerning the English editing:
Please make your sentences shorter. Too much information in one sentence produce only confusion. I noted several typing errors (ex: line 72: mono or oligosaccharides; fructose). Sentence line 102-107 is a good example for unclear formulation.
Concerning the background:
As usually, I have to complain about the amalgams that are done. line 51 to 60 have to be reconsider critically. It is not correct to make shortcuts from nutrition aspects to in vitro studies. So citations 15, 16 and 17 are abusive. 21-22 are not to be sorted from the "gut-pool" 19-20, especially by a sentence with "and so on"....
Concerning the work itself:
Major comments:
First concerning the plants, you are always talking about "g of plants", but it is important for the reader to know, how many individuals and areas are covered by your sampling. Are you studiing one place, one field, and one plant?
Secondly, I am quite surprised that 1 g plant diluted in 20 ml water allows to analyze monosacchrarides starting with 10µl injection volume on a semi-preparative column (1 mL/min) coupled too ELS detector. Did I miss something?
Finally, I do not agree the way you quantify the polysaccharides. The different sugars have very different response factors using colorimetric analysis. Therefore, deducing the abundancy of PS compared to glucose standard (without knowing their monosaccharidic composition) is doubtful. You have the technology allowing a better analysis, so use it. Analyze you hydrolysates with HPLC ELSD.
Minor remarks to correct:
Please define in the text line 120 and figure 2 what is sample S1 and why you choose to present it.
What is the meaning of "relatively wide concentration range" line 129?
In figure 3: the two first panels compare average values. Please indicate the error bars that are absolutely necessary to perform a comparison.
Figure 4 is not readable, please chose the most important data to present.
Author Response
Response to Reviewer 1 Comments
First I wanted to highlight the important work that have been done for this study.
Point 1: Concerning the English editing:
Please make your sentences shorter. Too much information in one sentence produce only confusion. I noted several typing errors (ex: line 72: mono or oligosaccharides; fructose). Sentence line 102-107 is a good example for unclear formulation.
Response 1: Thanks for your comments. We have modified the English language and make the sentences shorter to enhance readability.
Point 2: Concerning the background:
As usually, I have to complain about the amalgams that are done. line 51 to 60 have to be reconsider critically. It is not correct to make shortcuts from nutrition aspects to in vitro studies. So citations 15, 16 and 17 are abusive. 21-22 are not to be sorted from the "gut-pool" 19-20, especially by a sentence with "and so on"....
Response 2: Thanks for your comments. We have modified the paragraph “Introduction” to make a better and detailed description as follows: Trichosanthes kirilowii Maxim (T. kirilowii) is a perennial vine of the Cucurbitaceae family. The fruit, pericarp, seeds and roots of T. kirilowii are traditional Chinese medicines recorded in Chinese Pharmacopoeia 2015, named Trichosanthis tructus, Trichosanthis pericarpium, Trichosanthis semen and Trichosanthis radix, respectively. The fruit parts (fruit, pericarp and seeds) of T. kirilowii were mainly used for the treatment of cardiovascular diseases and lung diseases in the clinic. Besides, the root was rich in protein, which had cytotoxicity to tumor cells and hypoglycemic effects. In addition, the leaf, stem, fruit pulp and root bark of T. kirilowii were usually regarded as waste, took up valuable land resources and caused environmental problems. Studies have revealed that T. kirilowii contains various constituents such as terpenoids, flavonoids, phytosterols, amino acids, nucleosides, nucleobases and carbohydrates. However, pharmacological reports most often attribute some activities to various extracts rather than specific compounds. The major active compounds of T. kirilowii were not clear. Studies have shown that nucleosides, nucleobases and amino acids had clear pharmacological effects on the cardiovascular system. L-arginine was the substrate for endothelial nitric oxide synthesis, and modulated the development of atherosclerotic cardiovascular disease, improved immune function in healthy. L-citrulline supplementation increased nitric oxide synthesis, decreased blood pressure, and may increase peripheral blood flow. Cytosine and guanine had the effect of inhibiting platelet aggregation and improving endothelial vascular function. Meanwhile, carbohydrates also had extensive biological effects. Botanical polysaccharides from a wide array of different species of plant are a class of macromolecules that can markedly enhance and activate immune responses, leading to immunomodulation, antitumor activity, antiaging activity and other therapeutic effects.
Considering multiple bioactivities and benefits for human health of these compounds, several analytical methods to detect them have been reported to assess the quality of food or medicinal materials. However, nucleosides, nucleobases, amino acids and carbohydrates in medicinal organs (root, pericarp and seed kernels) and non-medicinal organs (fruit pulp, root bark, stem and leaf) of T. kirilowii had been rarely investigated. More information about the nutrients in different parts of T. kirilowii would, thus, have a significant impact on the efficient use of this valuable natural resource. Thus, it is necessary to establish a fast, convenient, and effective method to clearly characterize and quantify these constituents. On the basis of our previous research, the UHPLC coupled to triple quadrupole (TQ) MS - MS method to simultaneously detect and quantitate 14 nucleosides and 21 amino acids in different parts of T. kirilowii was developed and validated in this study. The high-performance liquid chromatography coupled with an evaporative light scattering detector (HPLC-ELSD) method was established for the detection of four monooligosaccharide, such as fructose, glucose, stachyose and raffinose. At the same time, polysaccharides in samples were determined by UV-visible (UV-Vis) spectrophotometer at 490 nm wavelength. Chemometrics are effective methods for multivariate statistical analysis, supervised partial least-squared discriminate analysis (PLS-DA) model and analysis of variance (ANOVA) were employed to find potential markers for the different organs (fruit, stems and leaves, roots) of T. kirilowii. The determination of nucleosides, amino acids and carbohydrates in the medicinal parts of T. kirilowii can provide theoretical support for the study of the efficacy of medicinal parts. At the same time, the determination of these nutrients can provide the development and utilization of non-medicinal organs.
Point 3: Concerning the work itself:
Major comments:
First concerning the plants, you are always talking about "g of plants", but it is important for the reader to know, how many individuals and areas are covered by your sampling. Are you studiing one place, one field, and one plant?
Response 3: Thanks for your comments. Samples of T. kirilowii were collected from the Breeding and Demonstration Base of Gualou Varieties in Changfeng County, Hefei, Anhui Province, China, in November 2017. Seven different planting varieties were collected: Wanlou 7 (S1), Wanlou 4-11 (S2), Wanlou 17 (S3), Wanlou 9 (S4), Wanlou 16 (S5), Wanlou 8 (S6), and Wanlou 7-8 (S7). The plants were authenticated by Prof. Dong Ling as T. kirilowii. After the samples were collected, the plants were divided into main root (A), lateral root (B), lateral root bark (C), main root bark (D), stem (E), leaf (F), pericarp (G), fruit pulp (H) and seed kernels (I). The pieces were freeze-dried and then pulverized to provide homogeneous powders (40 mesh) for extraction. We have modified the paragraph “3.2 Plant materials” to make a better and detailed description.
Point 4: Secondly, I am quite surprised that 1 g plant diluted in 20 ml water allows to analyze monosacchrarides starting with 10µl injection volume on a semi-preparative column (1 mL/min) coupled too ELS detector. Did I miss something?
Response 4: Thanks for your comments. Regarding the extraction method, the dried powder (1.0 g) of samples, which was accurately weighed, was put into a 50 mL glass-stoppered conical flask and 20 mL ultrapure water was added. After resting for one hour and accurate weighing, ultrasonication (40 kHz) was performed at room temperature for one hour. Afterwards solvent was added if there was any weight loss. After centrifugation (13,000 r/min, 10 min) and filtration (0.22 µm membrane filter), the supernatant was stored (4 °C) at a sample plate before injection into the HPLC systems for analysis. Some samples of pericarp and fruit pulp were diluted in appropriate proportion for determination of mono-oligosaccharide. In Chinese Pharmacopoeia, 1 g mel diluted in 20 ml solvent was used to analyze glucose and fructose starting with 10µL injection volume on a Prevail Carbohydrate ES column (1 mL/min) coupled with ELS detector [Chinese Pharmacopoeia Commission. Chinese Pharmacopoeia; China Medical Science Press: Beijing, China, 2015, 360.]. As to the established analytical method, the Prevail Carbohydrate ES (250 mm × 4.6 mm, 5 µm) column is a recognized analytical column for the determination of carbohydrates [1.Chinese Pharmacopoeia Commission. Chinese Pharmacopoeia; China Medical Science Press: Beijing, China, 2015, 360. 2. Zeng, H.; Su, S.; Xiang, X.; Sha, X.; Zhu, Z.; Wang, Y.; Guo, S.; Yan, H.; Qian, D.; Duan, J. Comparative Analysis of the Major Chemical Constituents in Salvia miltiorrhiza Roots, Stems, Leaves and Flowers during Different Growth Periods by UPLC-TQ-MS/MS and HPLC-ELSD Methods. Molecules. 2017, 22.]. As showed in paragraph “2.2. Method Validation”, the established HPLC-ELSD method was validated with good linearity, precision, repeatability, stability, recovery and was suitable for quantitative analysis. We have modified the statement to make a better and detailed description.
Point 5: Finally, I do not agree the way you quantify the polysaccharides. The different sugars have very different response factors using colorimetric analysis. Therefore, deducing the abundancy of PS compared to glucose standard (without knowing their monosaccharidic composition) is doubtful. You have the technology allowing a better analysis, so use it. Analyze you hydrolysates with HPLC ELSD.
Response 5: Thanks for your comments. The main purpose of this paper was to analyze the carbohydrates, nucleosides and amino acids in nine different parts of T. kirilowii and reveal the difference in chemical composition of different parts. The content of total polysaccharide is one of the indicators. As to determination of polysaccharide, according to the literature, the phenol and concentrated sulfuric acid method was a classic method for determining the polysaccharide content. In Chinese Pharmacopoeia [Chinese Pharmacopoeia Commission. Chinese Pharmacopoeia; China Medical Science Press: Beijing, China, 2015, 84+249+282.], the contents of polysaccharide in Lycii Fructus, Polygonati Odorati Rhizoma, Dendrobii Officinalis Caulis are determined determined by glucose as standard and phenol-sulfuric acid method. In addition, the content of polysaccharide in Chrysanthemum morifolium [He, J.; Chen, L.; Chu, B.; Zhang, C. Determination of Total Polysaccharides and Total Flavonoids in Chrysanthemum morifolium Using Near-Infrared Hyperspectral Imaging and Multivariate Analysis. Molecules. 2018, 23], Salvia miltiorrhiza [Xiang, X.; Sha, X.; Su, S.; Zhu, Z.; Guo, S.; Yan, H.; Qian, D.; Duan, J. A. Simultaneous determination of polysaccharides and 21 nucleosides and amino acids in different tissues of Salvia miltiorrhiza from different areas by UV-visible spectrophotometry and UHPLC with triple quadrupole MS/MS. J Sep Sci. 2018, 41, 996-1008.] and Ganoderma lingzhi [Nakagawa, T.; Zhu, Q.; Tamrakar, S.; Amen, Y.; Mori, Y.; Suhara, H.; Kaneko, S.; Kawashima, H.; Okuzono, K.; Inoue, Y.; Ohnuki, K.; Shimizu, K. Changes in content of triterpenoids and polysaccharides in Ganoderma lingzhi at different growth stages. J Nat Med. 2018, 72, 734-744.] were also determined by this method. We have modified the paragraph “3.5.3 Analysis for polysaccharides” to make a better and detailed description.
Point 6: Minor remarks to correct:
Please define in the text line 120 and figure 2 what is sample S1 and why you choose to present it.
Response 6: Thanks for your comments. As described in paragraph “3.2. Plant materials”, S1 represented Wanlou 7 and figure 2 presented chromatograms of mixed reference and leaves of sample S1 (Wanlou 7). The selection of S1 (Wanlou 7) is randomly, and the selection of leaves is because the leaves contain four mono-oligosaccharides of the measured carbohydrates, and the chromatogram was representative. Appropriate supplements were made in the manuscript with red words.
Point 7: What is the meaning of "relatively wide concentration range" line 129?
Response 7: Thanks for your comments. The original text was improperly expressed. We have modified the sentences to make a better and detailed description. All of the marker substances showed good linearity with the determination coefficients (R2) ranging from 0.9908 to 0.9999 within the determination ranges. Appropriate changes were made in the manuscript with red words.
Point 8: In figure 3: the two first panels compare average values. Please indicate the error bars that are absolutely necessary to perform a comparison.
Response 8: Thanks for your comments. We have added the error bars in the two first panels and replaced the original figure with a new one.
Point 9: Figure 4 is not readable, please chose the most important data to present.
Response 9: Thanks for your comments. The Figure 4 has been moved to the Supplemental material, Figure S2. And the proportion of the relatively abundant analytes in total amino acids was showed in Table 3.
Reviewer 2 Report
The authors of this MS have quantified carbohydrates, nucleosides, amino acids and polysaccharides in various parts of Trichosantes kirilowii Maxim., a plant usually employed in traditional Chinese medicine, by means of HPLC-ELSD, UV and UHPLC-MS/MS.
The MS is well written and organized. Contents are original and interesting.
Only few small revisions are needed prior publication:
- ll. 22-24: please revise this sentence. It seems that fructose, glucose, etc. are among amino acids.
- ll. 21,22 and 32: maybe it can be explained also in the abstract the meaning of acronyms used. For instance, ELSD and PLS-DA should not be known by non-expert readers
- l. 60: immunomodulators?
- l. 75: please indicate the meaning of the acronym PLS-DA
- Fig. 1: maybe this figure can be move in the supplementary material.
- Table 2: I should remove the column with regression equations. It sounds redundant.
- l. 312: The analytes were eluted
Author Response
Response to Reviewer 2 Comments
The authors of this MS have quantified carbohydrates, nucleosides, amino acids and polysaccharides in various parts of Trichosantes kirilowii Maxim., a plant usually employed in traditional Chinese medicine, by means of HPLC-ELSD, UV and UHPLC-MS/MS.
The MS is well written and organized. Contents are original and interesting.
Only few small revisions are needed prior publication:
Point 1: -Ⅱ. 22-24: please revise this sentence. It seems that fructose, glucose, etc. are among amino acids.
Response 1: Thanks for your comments. We have revised this sentence and polished the English languages for readability enhancement. In this study, high-performance liquid chromatography coupled with evaporative light scattering detector (HPLC-ELSD), UV-visible spectrophotometry and ultra-high-performance liquid chromatography coupled with tandem mass spectrometry (UHPLC-MS/MS) analytical methods for quantifying carbohydrates (fructose, glucose, stachyose, raffinose and polysaccharide), fourteen nucleosides and twenty one amino acids were established. Appropriate changes were made in the paragraph “Abstract”.
Point 2: -Ⅱ. 21,22 and 32: maybe it can be explained also in the abstract the meaning of acronyms used. For instance, ELSD and PLS-DA should not be known by non-expert readers
Response 2: Thanks for your comments. The explanations of abbreviations have been supplemented and marked red in the abstract.
Point 3: -Ⅰ. 60: immunomodulators?
Response 3: Thanks for your comments. The original text was improperly expressed. We have modified the words to make a better and detailed description.
Point 4: -Ⅰ. 75: please indicate the meaning of the acronym PLS-DA
Response 4: Thanks for your comments. The explanations of abbreviations have been supplemented and marked red in the text.
Point 5: - Fig. 1: maybe this figure can be move in the supplementary material.
Response 5: Thanks for your comments. In order to ensure the integrity of the analytical method, the authors believe that the Figure 1 could be in the main body.
Point 6: - Table 2: I should remove the column with regression equations. It sounds redundant.
Response 6: Thanks for your comments. The regression equations determined the extent to which the response was proportional to the concentration of the analyte in the sample. Regression equations were the basis for qualitative measurements. In order to ensure the integrity of the analytical method, the authors believe that the regression equation could be retained.
Point 7: -Ⅰ. 312: The analytes were eluted
Response 7: Thanks for your comments. “The analytes were eluted with a mixture of acetonitrile (mobile phase A) and water (mobile phase B) at a flow rate of 1.0 mL/min.” We have modified the English language and corrected the grammar mistakes to enhance readability.
Reviewer 3 Report
The manuscript entitled “Comparative analysis of carbohydrates, nucleosides and amino acids in different parts of Trichosanthes kirilowii Maxim. by HPLC-ELSD and UHPLC MS/MS methods” reported different analytical methods to quantify carbohydrates, nucleosides and amino acids in different parts of T. kirilowii (e.g., pericarp, seed, fruit pulp, stem, leaf) from different cultivated varieties. In my opinion, this manuscript is suitable for publication in Molecules only after a serious and exhaustive major revision. Although the manuscript has some positive components (the number of identified compounds, potential interest to the industry), it has particular weaknesses, that must be modified, namely:
1) The title should not contain abbreviations. Moreover, all abbreviations should be described for the first time.
2) In the introduction section, the authors wrote the following sentence “T. kirilowii has been rarely investigated, so that the highly efficient utilization of T. kirilowii resource is hindered. Thus, it is necessary to establish a fast, convenient, and effective method to clearly characterize and quantify these constituents.” So, what is the novelty of the present work? Several studies have been already published related to T. kirilowii composition as well as potential health benefits. I don’t understand the novelty.
3) The abstract is too long and important information is missing, such as “The established methods were reliable and sensitive. ”This sentence is based in which results? Moreover, the information related to different parts of T. kirilowii should be removed from this section in order to make the reading easier.
4) The results are insufficient explored. The authors should compare the composition of different parts of T. kirilowii, according previous studies explore the possible health benefits.
5) Line 72 rutose should be fructose
6) Table 2. The linear range should be expressed as ng/mL. Moreover, the authors should pay attention to significant numbers.
7) Line 169: please remove the double and
8) Line 189: crossvalidation parameter should be cross validation parameter
Author Response
Response to Reviewer 3 Comments
The manuscript entitled “Comparative analysis of carbohydrates, nucleosides and amino acids in different parts of Trichosanthes kirilowii Maxim. by HPLC-ELSD and UHPLC MS/MS methods” reported different analytical methods to quantify carbohydrates, nucleosides and amino acids in different parts of T. kirilowii (e.g., pericarp, seed, fruit pulp, stem, leaf) from different cultivated varieties. In my opinion, this manuscript is suitable for publication in Molecules only after a serious and exhaustive major revision. Although the manuscript has some positive components (the number of identified compounds, potential interest to the industry), it has particular weaknesses, that must be modified, namely:
Point 1: The title should not contain abbreviations. Moreover, all abbreviations should be described for the first time.
Response 1: Thanks for your comments. We have modified the title to “Comparative analysis of carbohydrates, nucleosides and amino acids in different parts of Trichosanthes kirilowii Maxim. by (ultra) high-performance liquid chromatography coupled with tandem mass spectrometry and evaporative light scattering detector methods”. All abbreviations have been described for the first time.
Point 2: In the introduction section, the authors wrote the following sentence “T. kirilowii has been rarely investigated, so that the highly efficient utilization of T. kirilowii resource is hindered. Thus, it is necessary to establish a fast, convenient, and effective method to clearly characterize and quantify these constituents.” So, what is the novelty of the present work? Several studies have been already published related to T. kirilowii composition as well as potential health benefits. I don’t understand the novelty.
Response 2: Thanks for your comments. We have modified the paragraph “Introduction” to make a better and detailed description. The fruit parts (fruit, pericarp and seeds) of T. kirilowii were mainly used for the treatment of cardiovascular diseases and lung diseases in the clinic. Besides, the root was rich in protein, which had cytotoxicity to tumor cells and hypoglycemic effects. In addition, the leaf, stem, fruit pulp and root bark of T. kirilowii were usually regarded as waste, took up valuable land resources and caused environmental problems. Studies have revealed that T. kirilowii contains various constituents such as terpenoids, flavonoids, phytosterols, amino acids, nucleosides, nucleobases and carbohydrates. However, pharmacological reports most often attribute some activities to various extracts rather than specific compounds. The major active compounds of T. kirilowii were not clear. Studies have shown that nucleosides, nucleobases and amino acids had clear pharmacological effects on the cardiovascular system. The determination of nucleosides, amino acids and carbohydrates in the medicinal parts of T. kirilowii can provide theoretical support for the study of the efficacy of medicinal parts. At the same time, the determination of these nutrients can provide the development and utilization of non-medicinal organs. However, there were hardly any references about thorough quantity analysis of carbohydrates, nucleosides and amino acids in Trichosanthes kirilowii. We searched in Pubmed with keywords “Trichosanthes” and “nucleosides” or “amino acids” or “carbohydrates”. For “Trichosanthes” and “nucleosides”, only one article was found, which introduced toxicity and activity of purified trichosanthin. Meanwhile, keywords “Trichosanthes” and “amino acids” /“carbohydrates”were also used for search. For “Trichosanthes” and “amino acids”, fifty seven articles were found, which were about isolation, identification and studying of amino acid sequences of proteins from roots or seeds, rather than the determination of amino acids. For “Trichosanthes” and “carbohydrates”, eighty articles were found, most of which were about isolation and purification of lectin, only two of which about extraction and activity of polysaccharides. Our paper was to determination of nucleosides, amino acids and carbohydrates in the medicinal parts and non-medicinal organs of T. kirilowii, thus, more useful information could be provided for the study and utilization of Trichosanthes kirilowii Maxim.
Search results are as follows:
Trichosanthes AND nucleosides: one article
[1] Ferrari P , Trabaud M A , Rommain, et al. Toxicity and activity of purified trichosanthin[J]. AIDS, 1991, 5(7):865-870.
Trichosanthes AND amino acids: fifty seven articles
[1] Purification and characterization of a novel chitinase from Trichosanthes dioica seed with antifungal activity[J]. International Journal of Biological Macromolecules, 2016, 84:62-68.
[2] Da-Hui L , Gui-Liang J , Ying-Tao Z , et al. Bacterial expression of aTrichosanthes kirilowiidefensin (TDEF1) and its antifungal activity onFusarium oxysporum[J]. APPLIED MICROBIOLOGY AND BIOTECHNOLOGY, 2007, 74(1):146-151.
[3] Kondo T , Mizukami H , Takeda T , et al. Amino Acid Sequences and Ribosome-Inactivating Activities of Karasurin-B and Karasurin-C.[J]. Biological & Pharmaceutical Bulletin, 1996, 19(11):1485-1489.
[4] Chow L P , Kamo M , Lin J Y , et al. Amino Acid Sequence ofTrichoanguina, a Ribosomal-Inactivating Protein fromTrichosanthes anguina Seeds[J]. Journal of Biomedical Science, 1996, 3(3):178-186.
[5] Cai X , Yao G , Xu G , et al. Identification of the amino acid residues in trichosanthin crucial for IgE response[J]. Biochemical and Biophysical Research Communications, 2002, 297(3):0-516.
[6] Ng T B , Chan W Y , Yeung H W . Proteins with abortifacient, ribosome inactivating, immunomodulatory, antitumor and anti-AIDS activities from Cucurbitaceae plants[J]. General Pharmacology the Vascular System, 1992, 23(4):0-590.
[7] Flores S H E. Biosynthesis of Defense-Related Proteins in Transformed Root Cultures of Trichosanthes kirilowii Maxim. var japonicum (Kitam.)[J]. Plant Physiology, 1994, 106(3):1195-1204.
[8] Weng IT, Lin YA. (-)-β-Homoarginine anhydride, a new antioxidant and tyrosinase inhibitor, and further active components from Trichosanthes truncata.[J]. Nat Prod Res, 2018, 23:1-7.
[9] Toshiya K,Satoko K. Effect of N- and C-terminal deletions on the RNAN-glycosidase activity and the antigenicity of karasurin-A, a ribosome-inactivating protein from Trichosanthes kirilowiivar.japonica[J]. Biotechnol Lett, 2004, 26(24): 1873-8.
[10] Kaneda M, Sobue A, Eida S, et al. Isolation and characterization of proteinases from the sarcocarp of snake-gourd fruit[J]. Journal of Biochemistry, 1986, 99(2):569.
[11] Toyokawa S, Takeda T, Kato Y, et al. The complete amino acid sequence of an abortifacient protein, karasurin[J]. Chemical & Pharmaceutical Bulletin, 2008, 39(5):1244-9.
[12] Yeung H W, W. W. L I. β-Trichosanthin: a new abortifacient protein from the Chinese drug, Wangua, Trichosanthes cucumeroides[J]. International Journal of Peptide & Protein Research, 1987, 29(3):289-292.
[13] Maraganore J M, Joseph M, Bailey M C. Purification and characterization of trichosanthin. Homology to the ricin A chain and implications as to mechanism of abortifacient activity.[J]. Journal of Biological Chemistry, 1987, 262.
[14] Dong T X , Ng T B , Yeung H W , et al. Isolation and characterization of a novel ribosome-inactivating protein, beta-kirilowin, from the seeds of Trichosanthes kirilowii.[J]. Biochemical & Biophysical Research Communications, 1994, 199(1):387-93.
[15] Synthesis and characterization of zinc sulfide quantum dots and their interaction with snake gourd (Trichosanthes anguina) seed lectin[J]. Spectrochimica Acta Part A: Molecular and Biomolecular Spectroscopy, 2015, 151:739-745.
[16] Yeung H W, T. B. N G , Wong D M , et al. Chemical and biological characterization of the galactose binding lectins from Trichosanthes kirilowii root tubers[J]. Chemical Biology & Drug Design, 1986, 27(2):208-220.
[17] Yeung H W , Poon S P , Ng T B , et al. Isolation and Characterization of An Immunosuppressive Protein From Trichosanthes Kirilowii Root Tubers[J]. Immunopharmacology and Immunotoxicology, 1987, 9(1):25-46.
[18] Srivastava S, Verma H , Srivastava A , et al. BDP-30, a systemic resistance inducer fromBoerhaavia diffusaL. suppresses TMV infection, and displays homology with ribosome-inactivating proteins[J]. Journal of Biosciences, 2015, 40(1):125-135.
[19] Lee-Huang S , Huang P L , Kung H F , et al. TAP 29: An Anti-Human Immunodeficiency Virus Protein from Trichosanthes kirilowii That is Nontoxic to Intact Cells[J]. Proceedings of the National Academy of Sciences of the United States of America, 1991, 88(15):6570-6574.
[20] Tan F L , Zhang G D , Mu J F , et al. Purification, Characterization and Sequence Determination of a Double-headed Trypsin Inhibitor Peptide from\r,Trichosanthes kirilowii\r, (a Chinese Medical Herb)[J]. Hoppe-Seyler′s Zeitschrift für physiologische Chemie, 1984, 365(2):1211-1218.
[21] Chen X M , Qian Y W , Chi C W , et al. Chemical Synthesis, Molecular Cloning, and Expression of the Gene Coding for the Trichosanthes Trypsin Inhibitor—a Squash Family Inhibitor1[J]. The Journal of Biochemistry, 1992, 112(1):45-51.
[22] Mizukami H , Iida K , Kondo T , et al. Cloning and Bacterial Expression of a Gene Encoding Ribosome-Inactivating PROTEINS, Karasurin-A and Karasurin-C, from Trichosanthes kirilowii var. Japonica.[J]. Biological & Pharmaceutical Bulletin, 1997, 20(6):711-713.
[23] Huang Z F, Wu M L , Qi Z W . Total synthesis of Trichosanthes trypsin inhibitor and its analogue[J]. Science in China. Series B, Chemistry, life sciences & earth sciences, 1990, 33(10):1192-1200.
[24] Iwabuchi M, Kohnomurase J , Imamura J . Delta 12-oleate desaturase-related enzymes associated with formation of conjugated trans-delta 11, cis-delta 13 double bonds.[J]. Journal of Biological Chemistry, 2003, 278(7):4603.
[25] Fei X, Hill M , Ma X , et al. Isolation of a putative ribosome inactivating protein from dried roots of Trichosanthes kirilowii used in Traditional Chinese Medicine.[J]. Planta Medica, 2004, 70(04):364-365.
[26] Chow L P, Chou M H , Ho C Y , et al. Purification, characterization and molecular cloning of trichoanguin, a novel type I ribosome-inactivating protein from the seeds of Trichosanthes anguina.[J]. Biochemical Journal, 1999, 338(1):211-219.
[27] Qian Y W, Tan F L , Qi Z W , et al. Studies on natural and modified peptide Trichosanthes trypsin inhibitors.[J]. Science in China. Series B, Chemistry, life sciences & earth sciences, 1990, 33(5):599-605.
[28] Sang M, Ying Y, Wu Q, et al. Cloning of a novel trypsin inhibitor from the Traditional Chinese medicine decoction pieces, Radix Trichosanthis.[J] Anal Biochem, 2019, doi: 10.1016/j.ab.2019.02.028.
[29] Fan H Y, Tao T , Dong S W , et al. Trichosanthes kirilowii : A New Host of Cucurbit mild mosaic virus in China[J]. Plant Disease, 2013, 97(10):1388-1388.
[30] Ling M H, Chi C W , Shao P Z . Cloning and Sequence Analyzing the Gene of Trichosanthes Trypsin Inhibitor.[J]. Sheng Wu Hua Xue Yu Sheng Wu Wu LI Xue Bao, 1996, 28(3):233-239.
[31] Falasca A I , Abbondanza A , Barbieri L , et al. Purification and partial characterization of a lectin from the seeds of Trichosanthes kirilowii Maximowicz.[J]. Febs Letters, 1989, 246(1):159-162.
[32] Krishnan R , Mcdonald K A , Dandekar A M , et al. Expression of recombinant trichosanthin, a ribosome-inactivating protein, in transgenic tobacco[J]. Journal of Biotechnology, 2002, 97(1):69-88.
[33] Yamashita K, Umetsu K, Suzuki T, et al. Purification and characterization of a Neu5Ac.alpha.2. fwdarw. 6Gal.beta.1. fwdarw. 4GlcNAc and HSO3-. fwdarw. 6Gal.beta.1. fwdarw. 4GlcNAc specific lectin in tuberous roots of Trichosanthes japonica[J]. Biochemistry, 1992, 31(46):11647-11650.
[34] Ke Y B, Chen J K , Nie H L , et al. Structure-function relationship of trichosanthin[J]. Life Sciences, 1997, 60(7):465-472.
[35] Wong R N, Dong T X , Ng T B , et al. alpha-Kirilowin, a novel ribosome-inactivating protein from seeds of Trichosanthes kirilowii (family Cucurbitaceae): a comparison with beta-kirilowin and other related proteins.[J]. Int J Pept Protein Res, 2010, 47(1-2):103-109.
[36] Chi P V , Truong H Q , Ha N T , et al. Characterization of trichobakin, a type I ribosome-inactivating protein from Trichosanthes sp. Bac Kan 8-98[J]. Biotechnology & Applied Biochemistry, 2001, 34(Pt 2):85.
[37] Lei H T, Song J J , Qi J J , et al. [Genetic transformation of hairy roots in Trichosanthes kirilowii Maxim. by Ti and Ri plasmids][J]. Zhongguo Zhong yao za zhi = Zhongguo zhongyao zazhi = China journal of Chinese materia medica, 2001, 26(3):162.
[38] Shih N J R, Mcdonald K A , Dandekar A M , et al. A novel type-1 ribosome-inactivating protein isolated from the supernatant of transformed suspension cultures of Trichosanthes kirilowii[J]. Plant Cell Reports, 1998, 17(6-7):531-537.
[39] Shivhare Y , Singour P K , Patil U K , et al. Wound healing potential of methanolic extract of Trichosanthes dioica Roxb (fruits) in rats[J]. Journal of ethnopharmacology, 2010, 127(3):0-619.
[40] Wang K Y, Xu Q. Lectins and Toxins[J]. 2000.
[41] Mei-Hing Y , Rong-Huan Z , Walter K K H , et al. Cloning of trichosanthin cDNA and its expression in Escherichia coli[J]. Gene, 1991, 97(2):267-272.
[42] Kumagai M H , Turpen T H , Weinzettl N , et al. Rapid, high-level expression of biologically active alpha-trichosanthin in transfected plants by an RNA viral vector.[J]. Proceedings of the National Academy of Sciences, 1993, 90(2):427-430.
[43] Hyuncheol O, Yeun-Ja M , Sook-Jung I , et al. Cucurbitacins from Trichosanthes kirilowii as the inhibitory components on tyrosinase activity and melanin synthesis of B16/F10 melanoma cells[J]. Planta Medica, 2002, 68(09):832-833.
[44] Jadão, A. S, Buriola J E , Rezende J A M . First report of Papaya ringspot virus-type W and Zucchini yellow mosaic virus infecting Trichosanthes cucumerina in Brazil.[J]. Plant Disease, 2010, 94(6):789-789.
[45] Maoka T, Hataya T . The Complete Nucleotide Sequence and Biotype Variability of Papaya leaf distortion mosaic virus [J]. Phytopathology, 2005, 95(2):128-135.
[46] Yeung H W, Li W W , Ng T B . Isolation of ribosome-inactivating and abortifacient protein from seeds of Luffa acutangula[J]. Chemical Biology & Drug Design, 2010, 38(1):15-19.
[47] Yamashita K, Umetsu K, Suzuki T, et al. Purification and characterization of a Neu5Ac.alpha.2. fwdarw. 6Gal.beta.1. fwdarw. 4GlcNAc and HSO3-. fwdarw. 6Gal.beta.1. fwdarw. 4GlcNAc specific lectin in tuberous roots of Trichosanthes japonica[J]. Biochemistry, 1992, 31(46):11647-11650.
[48] Komath S S, Nadimpalli S K , Swamy M J . Identification of histidine residues in the sugar binding site of snake gourd (Trichosanthes anguina) seed lectin.[J]. Iubmb Life, 2010, 44(1):107-116.
[49] Kabir S. The novel peptide composition of the seeds of Trichosanthes dioica Roxb.[J]. Cytobios, 2000, 103(403):121-31.
[50] Gan J H, Yu L, Wu J , et al. The Three-Dimensional Structure and X-Ray Sequence Reveal that Trichomaglin Is a Novel S-like Ribonuclease[J]. Structure (Cambridge), 2004, 12(6):1015-1025.
[51] Toyokawa S, Takeda T , Kato Y , et al. Presence of Protein Polymorphism in Karasurin, an Abortifacient and Anti-tumor Protein, Identified with Physicochemical Properties.[J]. CHEMICAL & PHARMACEUTICAL BULLETIN, 1991, 39(8):2132-2134.
[52] Keung W M , Yeung H W , Feng Z , et al. Importance of lysine and arginine residues to the biological activity of trichosanthin, a ribosome-inactivating protein from Trichosanthes kirilowii tubers[J]. International Journal of Peptide & Protein Research, 1993, 42(6):504-508.
[53] Yang L , Wu S , Zhang Q , et al. 23,24-Dihydrocucurbitacin B induces G2/M cell-cycle arrest and mitochondria-dependent apoptosis in human breast cancer cells (Bcap37)[J]. Cancer Letters, 2007, 256(2):0-278.
[54] Ferrari P , Trabaud M A , Rommain, Michèle, et al. Toxicity and activity of purified trichosanthin[J]. AIDS, 1991, 5(7):865-870.
[55] Hellings M , Maeyer M D , Verheyden S , et al. The Dead-End Elimination method, tryptophan rotamers, and fluorescence lifetimes[J]. Biophysical Journal, 2003, 85(3):1894-1902.
[56] Takemoto D J . Effect of trichosanthin an anti-leukemia protein on normal mouse spleen cells[J]. Anticancer Research, 1998, 18(1A):357.
[57] Zheng H , Wang B , Shaw P , et al. [Cloning and DNA sequencing of the gene encoding trichosanthin][J]. Acta Genetica Sinica, 1994, 21(1):42-51.
Trichosanthes AND carbohydrates: Eighty articles
[1] Zhang M , Su N , Huang Q , et al. Phosphorylation and antiaging activity of polysaccharide from, Trichosanthes, peel[J]. Journal of Food and Drug Analysis, 2017:S1021949817300376.
[2] Bamidele O P , Fasogbon M B . Chemical and antioxidant properties of snake tomato (Trichosanthes cucumerina) juice and Pineapple (Ananas comosus) juice blends and their changes during storage[J]. Food Chemistry, 2017, 220:184-189.
[3] Chen F , Li D , Shen H , et al. Polysaccharides from Trichosanthes Fructus via Ultrasound-Assisted Enzymatic Extraction Using Response Surface Methodology[J]. Biomed Research International, 2017, 2017(25):1-13.
[4] Kharbanda C , Alam M , Hamid H , et al. Ameliorative Effects of Trichosanthes dioica Extract in Suppressing Inflammatory Mediators and Attenuating Oxidative Stress[J]. Planta Medica, 2015, 81(05):348-356.
[5] Sharma A , Pohlentz G , Bobbili K B , et al. The sequence and structure of snake gourd (Trichosanthes anguina) seed lectin, a three-chain nontoxic homologue of type II RIPs[J]. Acta Crystallogr D Biol Crystallogr, 2013, 69(8):1493-1503.
[6] Qiong L I , Xiao-Li Y E , Hong Z , et al. Study on the Extraction Technology and Hypoglycemic Activity of Lectin from Trichosanthes kirilowi[J]. Journal of Chinese Medicinal Materials, 2012, 35(3):475-479.
[7] Narahari A , Nareddy P K , Swamy M J . A new chitooligosaccharide specific lectin from snake gourd (Trichosanthes anguina) phloem exudate. Purification, physico-chemical characterization and thermodynamics of saccharide binding[J]. Biochimie, 2011, 93(10):0-1684.
[8] Falasca A I , Abbondanza A , Barbieri L , et al. Purification and partial characterization of a lectin from the seeds of Trichosanthes kirilowii Maximowicz.[J]. Febs Letters, 1989, 246(1):159-162.
[9] Yao J , Nellas R B , Glover M M , et al. Stability and Sugar Recognition Ability of Ricin-like Carbohydrate Binding Domains[J]. Biochemistry, 2011, 50(19):4097-4104.
[10] Kanchanapoom T , Kasai R , Yamasaki K . Cucurbitane, hexanorcucurbitane and octanorcucurbitane glycosides from fruits of Trichosanthes tricuspidata[J]. Phytochemistry (Oxford), 2002, 59(2):215-228.
[11] Liu L , Bestel S , Shi J , et al. Paleolithic human exploitation of plant foods during the last glacial maximum in North China[J]. Proceedings of the National Academy of Sciences, 2013, 110(14):5380-5385.
[12] Kavitha M, Swamy MJ. Spectroscopic and differential scanning calorimetric studies on the unfolding of Trichosanthes dioicaseed lectin. Similar modes of thermal and chemical denaturation[J]. Glycoconj J, 2009, 26(8): 1075-84.
[13] Yamashita K, Umetsu K, Suzuki T, et al. Purification and characterization of a Neu5Ac.alpha.2. fwdarw. 6Gal.beta.1. fwdarw. 4GlcNAc and HSO3-. fwdarw. 6Gal.beta.1. fwdarw. 4GlcNAc specific lectin in tuberous roots of Trichosanthes japonica[J]. Biochemistry, 1992, 31(46):11647-11650.
[14] Hikino H , Yoshizawa M , Suzuki Y , et al. Isolation and Hypoglycemic Activity of Trichosans A, B, C, D, and E: Glycans of Trichosanthes kirilowii Roots 1[J]. Planta Medica, 1989, 55(4):349-350.
[15] Kitagawa T , Bai G , Fujiwara K , et al. A New Method for the Detection and Quantitative Measurement of the Contents of Trichosanthes Root Component Composing Chinese Traditional Medicines.[J]. Biological & Pharmaceutical Bulletin, 1996, 19(6):783-790.
[16] Wu A M , Wu J H , Tsai M S , et al. Carbohydrate specificity of an agglutinin isolated from the root of Richosanthes kirilowii[J]. Life Sciences, 2000, 66(26):2571-2581.
[17] Arawwawala M , Thabrew I , Arambewela L , et al. Anti-inflammatory activity of Trichosanthes cucumerina Linn. in rats[J]. Journal of Ethnopharmacology, 2010, 131(3):0-543.
[18] Wong C M , Yeung H W , Ng T B . Screening of Trichosanthes kirilowii, Momordica charantia and Cucurbita maxima (family Cucurbitaceae) for compounds with antilipolytic activity.[J]. Journal of Ethnopharmacology, 1985, 13(3):313-321.
[19] Jiandong L , Yang Y , Peng J , et al. Trichosanthes kirilowii lectin ameliorates streptozocin-induced kidney injury via modulation of the balance between M1/M2 phenotype macrophage[J]. Biomedicine & Pharmacotherapy, 2019, 109:93-102.
[20] Yeung H W , Poon S P , Ng T B , et al. Isolation and Characterization of An Immunosuppressive Protein From Trichosanthes Kirilowii Root Tubers[J]. Immunopharmacology and Immunotoxicology, 1987, 9(1):25-46.
[21] Kenoth R , Komath S S , Swamy M J . Physicochemical and saccharide-binding studies on the galactose-specific seed lectin from Trichosanthes cucumerina[J]. Archives of Biochemistry and Biophysics, 2003, 413(1):0-138.
[22] Srivastava S , Verma H , Srivastava A , et al. BDP-30, a systemic resistance inducer fromBoerhaavia diffusaL. suppresses TMV infection, and displays homology with ribosome-inactivating proteins[J]. Journal of Biosciences, 2015, 40(1):125-135.
[23] Choi C H , Kim T H , Sung Y K , et al. SKI306X inhibition of glycosaminoglycan degradation in human cartilage involves down-regulation of cytokine-induced catabolic genes[J]. The Korean Journal of Internal Medicine, 2014, 29(5):647-655.
[24] Sun X Y , Wu H H , Fu A Z , et al. [Chemical constituents of Trichosanthes kirilowii Maxim].[J]. Yao Xue Xue Bao, 2012, 47(7):922-925.
[25] Sultan N A M , Kavitha M , Swamy M J . Purification and physicochemical characterization of two galactose-specific isolectins from the seeds of Trichosanthes cordata[J]. International Union of Biochemistry and Molecular Biology Life, 2009, 61(4):457-469.
[26] Arawwawala L D A M , Thabrew M I , Arambewela L S R . Gastroprotective activity of Trichosanthes cucumerina in rats[J]. Journal of Ethnopharmacology, 2010, 127(3):0-754.
[27] Yamashita K, Umetsu K, Suzuki T, et al. Purification and characterization of a Neu5Ac.alpha.2. fwdarw. 6Gal.beta.1. fwdarw. 4GlcNAc and HSO3-. fwdarw. 6Gal.beta.1. fwdarw. 4GlcNAc specific lectin in tuberous roots of Trichosanthes japonica[J]. Biochemistry, 1992, 31(46):11647-11650.
[28] Komath S S , Kenoth R , Giribabu L , et al. Fluorescence and absorption spectroscopic studies on the interaction of porphyrins with snake gourd (Trichosanthes anguina) seed lectin[J]. Journal of Photochemistry and Photobiology B Biology, 2000, 55(1):49-55.
[29] Stoner M R , Humphrey C A , Coutts D J , et al. Kinetics of Growth and Ribosome-Inactivating Protein Production from Trichosanthes kirilowii Plant Cell Cultures in a 5-L Bioreactor[J]. Biotechnology Progress, 2010, 13(6):799-804.
[30] Ito H , Hoshi K , Osuka F , et al. “Lectin inhibits antigen-antibody reaction in a glycoform-specific manner: Application for detecting α2,6sialylated-carcinoembryonic antigen”[J]. PROTEOMICS, 2016.
[31] Yeung H W , T. B. N G , Wong D M , et al. Chemical and biological characterization of the galactose binding lectins from Trichosanthes kirilowii root tubers[J]. Chemical Biology & Drug Design, 1986, 27(2):208-220.
[32] Ai-Feng L I , Ai-Ling S , Ren-Min L , et al. Chemical Constituents of Trichosanthes kirilowii Peels[J]. Journal of Chinese Medicinal Materials, 2014, 37(3):428-431.
[33] Yeung H W , W. W. L I . β-Trichosanthin: a new abortifacient protein from the Chinese drug, Wangua, Trichosanthes cucumeroides[J]. International Journal of Peptide & Protein Research, 1987, 29(3):289-292.
[34] Bhattacharya S , Haldar P K . Protective role of the triterpenoid-enriched extract of Trichosanthes dioica root against experimentally induced pain and inflammation in rodents[J]. Natural Product Letters, 2012, 26(24):5.
[35] Sultan N A M , Kenoth R , Swamy M J . Purification, physicochemical characterization, saccharide specificity, and chemical modification of a Gal/GalNAc specific lectin from the seeds of Trichosanthes dioica[J]. Archives of Biochemistry & Biophysics, 2004, 432(2):0-221.
[36] Komath S S , Kenoth R , Swamy M J . Thermodynamic analysis of saccharide binding to snake gourd (Trichosanthes anguina) seed lectin. Fluorescence and absorption spectroscopic studies.[J]. European Journal of Biochemistry, 2010, 268(1):111-119.
[37] Fan X M , Chen G , Sha Y , et al. Chemical constituents from the fruits of Trichosanthes kirilowii[J]. Journal of Asian natural products research, 2012, 14(6):528-532.
[38] Yan J , Wang Y , Zhang X , et al. Snakegourd root/Astragalus polysaccharide hydrogel preparation and application in 3D printing[J]. International Journal of Biological Macromolecules, 2019, 121:309-316.
[39] Casellas P , Dussossoy D , Falasca A I , et al. Trichokirin, a ribosome-inactivating protein from the seeds of Trichosanthes kirilowii Maximowicz. Purification, partial characterization and use for preparation of immunotoxins[J]. Febs Journal, 2010, 176(3):581-588.
[40] Survey of glycoantigens in cells from α1-3galactosyltransferase knockout pig using a lectin microarray[J]. Xenotransplantation, 2010, 17(1):61-70.
[41] Lei H T , Song J J , Qi J J , et al. [Genetic transformation of hairy roots in Trichosanthes kirilowii Maxim. by Ti and Ri plasmids][J]. Zhongguo Zhong yao za zhi = Zhongguo zhongyao zazhi = China journal of Chinese materia medica, 2001, 26(3):162.
[42] Komath S S , Nadimpalli S K , Swamy M J . Purification in high yield and characterisation of the galactose-specific lectin from the seeds of snake gourd (Trichosanthes anguina).[J]. Biochem Mol Biol Int, 2010, 39(2):243-252.
[43] Song Y , Ding N , Kanazawa T , et al. Cucurbitacin D is a new inflammasome activator in macrophages[J]. International Immunopharmacology, 2013, 17(4):1044-1050.
[44] Nishijima Y , Toyoda M , Yamazaki-Inoue M , et al. Glycan profiling of endometrial cancers using lectin microarray[J]. Genes to Cells, 2012, 17(10):826-836.
[45] Lee G I , Ha J Y , Min K R , et al. Inhibitory effects of Oriental herbal medicines on IL-8 induction in lipopolysaccharide-activated rat macrophages[J]. Planta Medica, 1995, 61(01):26-30.
[46] Ng T B , Li W W , Yeung H W . Effects of ginsenosides, lectins and Momordica charantia insulin-like peptide on corticosterone production by isolated rat adrenal cells[J]. Journal of Ethnopharmacology, 1987, 21(1):21-29.
[47] Yoshikawa K , Umetsu K , Shinzawa H , et al. Determination of carbohydrate-deficient transferrin separated by lectin affinity chromatography for detecting chronic alcohol abuse[J]. FEBS Letters, 1999, 458(2):112-116.
[48] Yamashita K , , Fukushima K , , Sakiyama T , , et al. Expression of Sia alpha 2-->6Gal beta 1-->4GlcNAc residues on sugar chains of glycoproteins including carcinoembryonic antigens in human colon adenocarcinoma: applications of Trichosanthes japonica agglutinin I for early diagnosis[J]. Cancer Research, 1995, 55(8):1675-1679.
[49] Fukushima K , Satoh T , Baba S , et al. {alpha1,2-Fucosylated and {beta-N-acetylgalactosaminylated prostate-specific antigen as an efficient marker of prostatic cancer[J]. Glycobiology, 2010, 20(4):452-460.
[50] Mozingo NM, Hedrick JL. Distribution of lectin binding sites in Xenopus laevis egg jelly.[J] Dev Biol, 1999, 210(2):428-39.
[51] Kar A , Choudhary B K , Bandyopadhyay N G . Comparative evaluation of hypoglycaemic activity of some Indian medicinal plants in alloxan diabetic rats[J]. Journal of Ethnopharmacology, 2003, 84(1):105-108.
[52] Xu W , Hou W , Yao G , et al. Inhibition of Th1- and Enhancement of Th2-Initiating Cytokines and Chemokines in Trichosanthin- Treated Macrophages[J]. Biochemical & Biophysical Research Communications, 2001, 284(1):0-172.
[53] Choi J H , Choi J H , Kim D Y , et al. Effects of SKI 306X, a new herbal agent, on proteoglycan degradation in cartilage explant culture and collagenase-induced rabbit osteoarthritis model[J]. Osteoarthritis Cartilage, 2002, 10(6):471-478.
[54] Yang X X , Li F , Hu W G , et al. Preparation and Preliminary Application of Monoclonal Antibodies against Trichokirin-S1, a Small Ribosome-inactivating Peptide from the Seeds of Trichosanthes kirilowii[J]. Acta Biochimica et Biophysica Sinica, 2005, 37(7):447-452.
[55] Machawal L , Singh S , Chauhan M . Trichosanthes dioica Roxb.: Pharmacognostic standardization of the female leaves with special emphasis on the microscopic technique[J]. Journal of Pharmacy and Bioallied Sciences, 2011, 3(2):249.
[56] Huang Y , He P , Bader K P , et al. Seeds of Trichosanthes kirilowii, an Energy-Rich Diet[J]. Zeitschrift für Naturforschung C, 2000, 55(3-4).
[57] Lian L , Fan X M , Chen G , et al. Two new compounds from the fruits of Trichosanthes kirilowii Maxim[J]. Journal of Asian Natural Products Research, 2012, 14(1):4.
[58] Saha S S , Ghosh M . Antioxidant and anti-inflammatory effect of conjugated linolenic acid isomers against streptozotocin-induced diabetes[J]. British Journal of Nutrition, 2012, 108(06):974-983.
[59] Shaw P C , Lee K M , Wong K B . Recent advances in trichosanthin, a ribosome-inactivating protein with multiple pharmacological properties[J]. Toxicon, 2005, 45(6):0-689.
[60] Chao Z, He B. Studies on chemical constituents from fruits of Trichosanthes kirilowii Maxim.[J] Zhongguo Zhong Yao Za Zhi, 1999, 24(10): 612-13.
[61] Hettiaratchi U P K , Ekanayake S , Welihinda J . Sri Lankan rice mixed meals: effect on glycaemic index and contribution to daily dietary fibre requirement.[J]. Malays J Nutr, 2011, 17(1):97-104.
[62] Anuradha P , Bhide S V . An isolectin complex from Trichosanthes anguina seeds[J]. Phytochemistry (Oxford), 1999, 52(5):751-758.
[63] Schalla W O , Rice R J , Biddle J W , et al. Lectin characterization of gonococci from an outbreak caused by penicillin-resistant Neisseria gonorrhoeae.[J]. Journal of Clinical Microbiology, 1985, 22(4):481-3.
[64] Cao H L , Sun L H , Li J , et al. A quality comparison of protein crystals grown under containerless conditions generated by diamagnetic levitation, silicone oil and agarose gel.[J]. Acta Crystallographica, 2013, 69(12):2583-2583.
[65] Sharma G, Pant M C. Effect of raw deseeded fruit powder of Trichosanthes dioica (Roxb) on blood sugar, serum cholesterol, high density lipo-protein, phospholipid and triglyceride levels in the normal albino rabbits.[J]. Indian Journal of Physiology & Pharmacology, 1988, 32(2):161.
[66] Blanchette C D , Lin W C , Ratto T V , et al. Galactosylceramide Domain Microstructure: Impact of Cholesterol and Nucleation/Growth Conditions[J]. Biophysical Journal, 2006, 90(12):4466-4478.
[67] Cao L , Xu Y , Xu S , et al. [Effects of snakegourd root polysaccharide on apoptosis of MCF-7 cells].[J]. Journal of Zhejiang University, 2012, 41(5):527-534.
[68] Schalla W O , Whittington W L , Rice R J , et al. Epidemiological characterization of Neisseria gonorrhoeae by lectins[J]. Journal of Clinical Microbiology, 1985, 22(3):379-382.
[69] Hartog A , Hougee S , Faber J , et al. The multicomponent phytopharmaceutical SKI306X inhibits in vitro cartilage degradation and the production of inflammatory mediators[J]. Phytomedicine International Journal of Phytotherapy & Phytopharmacology, 2008, 15(5):313-320.
[70] Mai L P , Daniel Guénard, Franck M , et al. New Cytotoxic Cucurbitacins from the Pericarps of Trichosanthes Tricuspidata Fruits[J]. Natural Product Letters, 2002, 16(1):5.
[71] Seshagirirao K , Leelavathi C , Sasidhar V . Cross-linked Leucaena Seed Gum Matrix: An Affinity Chromatography Tool for Galactose-specific Lectins[J]. Journal of biochemistry and molecular biology, 2005, 38(3):370-372.
[72] Abdi K , Kobzik L , Li X , et al. Membrane glycoconjugates expressed on sheep airway epithelium.[J]. Journal of Histochemistry & Cytochemistry, 1994, 42(10):1341-1347.
[73] He X J , Wang N L , Qiu F , et al. [Study on the active spirostanol saponins of Gualou xiebai baijiutang][J]. Acta Pharmaceutica Sinica, 2003, 38(6):433-437.
[74] Yeung H W , Li W W , Ng T B . Isolation of ribosome-inactivating and abortifacient protein from seeds of Luffa acutangula[J]. Chemical Biology & Drug Design, 2010, 38(1):15-19.
[75] Hemmerich S , Rosen S D . 6\"-Sulfated Sialyl Lewis x Is a Major Capping Group of GlyCAM-1[J]. Biochemistry, 1994, 33(16):4830-4835.
[76] Komath S S , Nadimpalli S K , Swamy M J . Identification of histidine residues in the sugar binding site of snake gourd (Trichosanthes anguina) seed lectin.[J]. Iubmb Life, 2010, 44(1):107-116.
[77] He XJ, Qiu F, Shoyama Y, The active constituents from Gualou-xiebai-baijiu-tang part I: active saponins.[J] J Asian Nat Prod Res, 2002, 4(3): 189-96.
[78] Jiu H X , Li W N , Feng Q , et al. Research on Active Constituents Research of Gualou xiebai baijiutang (Ⅲ) the Active Flavanoids[J]. China Journal of Chinese Materia Medica, 2003.
[79] Sharma G , Pant M C . Preliminary observations on serum biochemical parameters of albino rabbits fed on seeds of Trichosanthes dioica (Roxb).[J]. Indian Journal of Medical Research, 1988, 87(2):398-400.
[80] Wang Y , Han S , Sun G , et al. [Experimental study on the hypoglycemic effect of tangniaole capsule].[J]. Journal of Chinese Medicinal Materials, 2002, 25(6):426.
Point 3: The abstract is too long and important information is missing, such as “The established methods were reliable and sensitive. ”This sentence is based in which results? Moreover, the information related to different parts of T. kirilowii should be removed from this section in order to make the reading easier.
Response 3: Thanks for your comments. We have changed the general description of “The established methods were reliable and sensitive.” into “The established methods were validated with good linearity, precision, repeatability, stability, and recovery.” Moreover, the detailed information related to characteristic components of different parts has been removed. Appropriate changes were made in the paragraph “Abstract” as follows: “Trichosanthes kirilowii Maxim. is one of the original plants for traditional Chinese medicines Trichosanthis Fructus, Trichosanthis Semen, Trichosanthis Pericarpium and Trichosanthis Radix. Amino acids, nucleosides and carbohydrates are usually considered to have nutritional value and health-care efficacy. In this study, high-performance liquid chromatography coupled with evaporative light scattering detector (HPLC-ELSD), UV-visible spectrophotometry and ultra-high-performance liquid chromatography coupled with tandem mass spectrometry (UHPLC-MS/MS) analytical methods for quantifying carbohydrates (fructose, glucose, stachyose, raffinose and polysaccharide), fourteen nucleosides and twenty one amino acids were established. Moreover, sixty-three samples from nine different parts, including pericarp, seed, fruit pulp, stem, leaf, main root, main root bark, lateral root and lateral root bark of T. kirilowii from different cultivated varieties were determined. The established methods were validated with good linearity, precision, repeatability, stability, and recovery. The results showed that the average content of total amino acids in roots (15.39 mg/g) and root barks (16.38 mg/g) were relatively higher than others. Contents of nucleosides in all parts of T. kirilowii were below 1.5 mg/g. For carbohydrates, fruit pulp has a higher content than others, with glucose (22.91%), fructose (20.63%) and polysaccharides (27.29%). By partial least-squared discriminate analysis (PLS-DA), Variable importance in the projection (VIP) plots and analysis of variance (ANOVA) analysis, the characteristic components of the different organs (fruit, stems and leaves, roots) were found. This analysis suggested there were potential medicinal and nutritive health care values in various parts of the T. kirilowii, which provided valuable information for the development and utilization of T. kirilowii.”
Point 4: The results are insufficient explored. The authors should compare the composition of different parts of T. kirilowii, according previous studies explore the possible health benefits.
Response 4: Thanks for your comments. As shown in “Quantitative analysis of samples”, the composition of different parts of T. kirilowii have been compared. A discussion about possible health benefits of different parts has been supplied. The established UHPLC-MS/MS method was subsequently applied to analyze 14 nucleosides and 21 amino acids in different parts of T. kirilowii. The concentrations of each analyte were calculated via an external standard method. As shown in Supplementary Materials Table S1, almost all of the T. kirilowii samples were rich in amino acids, despite their contents were obviously various, and the total content of amino acids varied from 0.4449 to 21.29 mg/g among different parts of T. kirilowii. As shown in Figure 3, the average content of total amino acids in roots (15.39 mg/g) and root barks (16.38 mg/g) were relatively higher than others, and in the order: lateral root barks (17.51 mg/g) > lateral roots (16.15 mg/g) > main root barks (15.27 mg/g) > main roots (14.64 mg/g) > pericarp (7.637 mg/g) > stem (7.177 mg/g) > leaves (7.003 mg/g) > fruit pulp (4.854 mg/g) > seed kernel (2.409 mg/g). The leaves was found to be the most abundant essential amino acid in all parts, and its average content in these investigated samples was 2.551 mg/g, which accounted for more than 35% of total amino acids tested in this study. Next was fruit pulp, whose average essential amino acid content was 1.784 mg/g. In terms of individual compound, as showed in Table 3 and Supplementary Materials figure S2, citrulline was found to be the most abundant in all parts except for fruit pulp and seed kernel, and the contents in roots and root barks were more than 9.2 mg/g, accounted for more than 60% of total amino acids.
Contents of nucleosides in all parts of T. kirilowii were below 1.5 mg/g, and only the contents in leaves (1.312 mg/g), lateral root barks (1.255 mg/g) and stem (1.054 mg/g) were above 1mg/g. Next, chemometrics, as effective tools of multivariate statistical analysis, were employed to classify and depict the intrinsic differences of T. kirilowii samples.
The results showed that the content of amino acids in the root was higher than other parts, so roots of T. kirilowii were good choice of raw material source for amino acids. The amino acid composition and content in the non-medicinal part root bark were similar to those in the root, thus, the non-medicinal part root bark could be a good choice of utilization of amino acids. In addition, the pericarp was also rich in amino acids, especially the citrulline. Citrulline increased nitric oxide synthesis and decreased blood pressure, which may be one of the material bases of pericarp for the treatment of cardiovascular diseases. The leaves of T. kirilowii were rich in essential amino acids, suggesting that the leaves could be developed into health products.
The HPLC-ELSD method was applied for mono-oligosaccharides analysis of the samples, and the contents of the four saccharides (fructose, glucose, raffinose and stachyose) were shown in Figure 3, which revealed that the contents of four saccharides in each part of T. kirilowii differ from each other. The contents of maltose, mannose and sucrose in all samples were below the limit of detection. In the fruit pulp of T. kirilowii, the content of glucose (22.91%), which possessed a high development value, was significantly higher than other parts of the plant, followed by the pericarp (21.77%). Similarly, the fructose was detected mostly in the pericarp (21.04%), followed by the fruit pulp (20.63%). However, the stachyose mainly exist in lateral roots and root barks. Raffinose mainly existed in leaves (1.32%). In addition, the results showed that the polysaccharides in fruit pulp reached up to 27.29%.
The results revealed that in T. kirilowii, the monosaccharides and polysaccharides probably mainly synthetize in genital organs, like fruit pulp and pericarp, while constituents of oligosaccharide may form by dehydration with different kinds of monosaccharides in conducting tissue and vegetative organs, such as stems and roots. Therefore, it can provide guidance for obtaining various saccharide resources from T. kirilowii, as was known to us that saccharides and their derivatives may always be the important components that are responsible for the pharmacological activities of traditional Chinese medicine (TCM). As a non-medicinal part, fruit pulp contained a large amount of fructose, glucose and polysaccharide, suggesting that fruit pulp of T. kirilowii might provide a supplementary source of saccharides.
Point 5: Line 72 rutose should be fructose
Response 5: Thanks for your comments. We have reviewed the manuscript and corrected those misspellings, and polished the English languages for readability enhancement.
Point 6: Table 2. The linear range should be expressed as ng/mL. Moreover, the authors should pay attention to significant numbers.
Response 6: Thanks for your suggestions. We have changed the unit of linear range to ng/mL and have checked significant numbers.
Point 7: Line 169: please remove the double and
Response 7: Thanks for your comments. We have corrected the mistake.
Point 8: Line 189: crossvalidation parameter should be cross validation parameter
Response 8: Thanks for your comments. We have corrected the mistake.
Round 2
Reviewer 1 Report
Thank you for this manuscript emprovment.
However, some points are still to adress:
The introductory sentence (lines 23-28) is not understandable. At least one verb is missing.
The sentence (lines 174-176) on citruline is really speculative, especially since this aspect has not been looked at in the present work. So please erase the sence.
In the experimental part, I asked for "what amount of sample (which volume and concentration) was analyzed by ELSD". The answer "appropriated proportions" does not satisfy me. Saying that is saying nothing at all.
Finally, I regret that my remark concerning the quantification of polysaccharides have not been taken into account. I persist: quantifying them by a simple colorimetric test (considering all the biases that it creates) when an ELS is available amazes me.
Author Response
Response to Reviewer 1 Comments
Thank you for this manuscript emprovment.
However, some points are still to adress:
Point 1: The introductory sentence (lines 23-28) is not understandable. At least one verb is missing.
Response 1: Thanks for your comments. We have modified the statement to make a better and detailed description. In this study, high-performance liquid chromatography coupled with evaporative light scattering detector (HPLC-ELSD), UV-visible spectrophotometry and ultra-high-performance liquid chromatography coupled with tandem mass spectrometry (UHPLC-MS/MS) methods were established for quantifying carbohydrates (fructose, glucose, stachyose, raffinose and polysaccharide), fourteen nucleosides and twenty one amino acids.
Point 2: The sentence (lines 174-176) on citruline is really speculative, especially since this aspect has not been looked at in the present work. So please erase the sence.
Response 2: Thanks for your comments. The sentence (lines 174-176) on citruline has been erased.
Point 3: In the experimental part, I asked for "what amount of sample (which volume and concentration) was analyzed by ELSD". The answer "appropriated proportions" does not satisfy me. Saying that is saying nothing at all.
Response 3: Thanks for your comments. The samples of pericarp and fruit pulp were diluted ten times with ultrapure water for determination of mono-oligosaccharide. And the injection volume was 10 µL for analysis. We have modified the statement to make a better and detailed description.
Point 4: Finally, I regret that my remark concerning the quantification of polysaccharides have not been taken into account. I persist: quantifying them by a simple colorimetric test (considering all the biases that it creates) when an ELS is available amazes me.
Response 4: Thanks for your comments. Your comments are very reasonable. To our knowledge, direct quantification of polysaccharides by ELSD is a new analytical method. It is reported that HPLC-ELSD method is used for monosaccharide composition analysis of polysaccharides. The existing method of quantifying polysaccharides is a colorimetric method (the phenol and concentrated sulfuric acid method). As to determination of polysaccharide, according to the literature, the phenol and concentrated sulfuric acid method was a classic method for determining the polysaccharide content. In Chinese Pharmacopoeia [Chinese Pharmacopoeia Commission. Chinese Pharmacopoeia; China Medical Science Press: Beijing, China, 2015, 84+249+282.], the contents of polysaccharide in Lycii Fructus, Polygonati Odorati Rhizoma, Dendrobii Officinalis Caulis are determined determined by glucose as standard and phenol-sulfuric acid method. In addition, the content of polysaccharide in Chrysanthemum morifolium [He, J.; Chen, L.; Chu, B.; Zhang, C. Determination of Total Polysaccharides and Total Flavonoids in Chrysanthemum morifolium Using Near-Infrared Hyperspectral Imaging and Multivariate Analysis. Molecules. 2018, 23], Salvia miltiorrhiza [Xiang, X.; Sha, X.; Su, S.; Zhu, Z.; Guo, S.; Yan, H.; Qian, D.; Duan, J. A. Simultaneous determination of polysaccharides and 21 nucleosides and amino acids in different tissues of Salvia miltiorrhiza from different areas by UV-visible spectrophotometry and UHPLC with triple quadrupole MS/MS. J Sep Sci. 2018, 41, 996-1008.] and Ganoderma lingzhi [Nakagawa, T.; Zhu, Q.; Tamrakar, S.; Amen, Y.; Mori, Y.; Suhara, H.; Kaneko, S.; Kawashima, H.; Okuzono, K.; Inoue, Y.; Ohnuki, K.; Shimizu, K. Changes in content of triterpenoids and polysaccharides in Ganoderma lingzhi at different growth stages. J Nat Med. 2018, 72, 734-744.] were also determined by this method. The method for directly determining the content of polysaccharide by HPLC-ELSD method mentioned by the reviewer is to be explored in our subsequent research.
Reviewer 3 Report
The manuscript entitled "Comparative analysis of carbohydrates, nucleosides and amino acids in different parts of Trichosanthes kirilowii Maxim. by (ultra) high-performance liquid chromatography coupled with tandem mass spectrometry and evaporative light scattering detector methods" should be accepted in present form.
Author Response
Response to Reviewer 3 Comments
The manuscript entitled "Comparative analysis of carbohydrates, nucleosides and amino acids in different parts of Trichosanthes kirilowii Maxim. by (ultra) high-performance liquid chromatography coupled with tandem mass spectrometry and evaporative light scattering detector methods" should be accepted in present form.
Thanks for your comments. I am very happy with your recognition of the paper named "Comparative analysis of carbohydrates, nucleosides and amino acids in different parts of Trichosanthes kirilowii Maxim. by (ultra) high-performance liquid chromatography coupled with tandem mass spectrometry and evaporative light scattering detector methods".